# Regionalization in global hydrological models and its impact on runoff simulations: A case study using WaterGAP3 (v 1.0.0)

Jenny Kupzig[1], Nina Kupzig[2], Martina Flörke[1]

[1]Institute of Engineering Hydrology and Water Resources Management, Ruhr-University, 44801, Bochum, Germany

[2]Faculty of Management and Economics, Ruhr-University, 44780, Bochum, Germany

*Correspondence to*: Jenny Kupzig (jenny.kupzig@rub.de)

**Abstract:**

Valid simulation results from global hydrological models (GHMs), such as WaterGAP3, are essential to detecting hotspots or studying patterns in climate change impacts. However, the lack of worldwide monitoring data makes it challenging to adapt GHMs' parameters to enable such valid simulations globally. Therefore, regionalization is necessary to estimate parameters in ungauged basins. This study presents the results of regionalization methods for the first time applied on the GHM WaterGAP3. It aims to provide insights into (1) selecting a suitable regionalization method for a GHM and (2) evaluating its impact on runoff simulation. In this study, four new regionalization methods have been identified as appropriate for WaterGAP3. These methods span the full spectrum of methodologies, i.e., regression-based methods, physical similarity, and spatial proximity, using traditional and machine learning-based approaches. Moreover, the methods differ in the descriptors used to achieve optimal results, although all utilize climatic and physiographic descriptors. This demonstrates (1) that different methods use descriptor sets with varying efficiency and (2) that combining climatic and physiographic descriptors is optimal for regionalizing worldwide basins. Additionally, our research indicates that regionalization leads to spatially and temporally varying uncertainty in ungauged regions. For example, regionalization highly affects southern South America, e.g., leading to high uncertainties in the flood simulation of the Río Deseado. The local impact of regionalization propagates through the water system, also affecting global estimates, as evidenced by a spread of 1,500 km³ yr⁻¹ across an ensemble of five regionalization methods in simulated global runoff to the ocean. This discrepancy is even more pronounced when using a regionalization method deemed unsuitable for WaterGAP3, resulting in a spread of 4,208 km³ yr⁻¹. This significant increase highlights the importance of carefully choosing regionalization methods. Further research is needed to enhance the predictor selection and the understanding of the methods' robustness on a global scale.

## 1. Introduction

Global hydrological models (GHMs) are developed and applied worldwide, e.g., to detect hotspots and examine patterns of climate change impacts on the terrestrial water cycle (e.g., Barbarossa et al., 2021; Boulange et al., 2021). Valid model results are a prerequisite to draw robust conclusions. For valid modeling results, it is beneficial to adjust the parameter values to adapt the models to different basin processes (Gupta et al., 1998). This adaptation is usually modified and evaluated (in a loop) by comparing the simulated model output, often discharge, with the monitored data. However, this parameter adjustment for GHMs is challenging due to the lack of global monitoring

data. Consequently, parameter adjustment for GHMs can be based not only on monitored data (i.e., calibration) but also on estimating parameter values for ungauged basins (i.e., regionalization).

Regionalization defines the estimation of model parameters for ungauged basins (Oudin et al., 2008), usually based on information from gauged basins (Oudin et al., 2010). Regionalization methods generally follow the same principle: basin characteristics (e.g., physiographic and/or climatic) are linked to hydrological characteristics and can thus be used to estimate parameter values. Various regionalization methods exist, and no overall preferred method has been found (Ayzel et al., 2017; Pool et al., 2021). In contrast, the optimal regionalization method may differ, for example, regarding available information (Pagliero et al., 2019) or model structures (Golian et al., 2021). Therefore, different methods should be tested to find an optimal regionalization method for a specific use case (e.g., Qi et al., 2020).

Evaluation is needed to assess different regionalization methods. The evaluation of regionalization methods is particularly challenging because they are usually applied when there is a lack of monitoring data. Therefore, regionalization studies often treat gauged basins as "ungauged" and perform leave-one-out cross-validation (e.g., Chaney et al., 2016) or split-sample tests (e.g., Beck et al., 2016; Nijssen et al., 2000; Yoshida et al., 2022). While at the mesoscale, this evaluation is already an integral part (e.g., McIntyre et al., 2005; Parajka et al., 2005; Oudin et al., 2008; Yang et al., 2020), this is sometimes not the case in global or continental studies (e.g., Müller Schmied et al., 2021; Widén-Nilsson et al., 2007). Another reasonable evaluation strategy is the concept of benchmark-to-beat (Schaefli & Gupta, 2007; Seibert, 2001). Applying a benchmark-to-beat supports a comprehensive evaluation of whether a new approach is functional, e.g., better than a straightforward and thus transparent method or better than a predecessor. To the authors' knowledge, such a benchmark-to-beat has never been used to evaluate innovations in regionalization at a global scale.

In general, regionalization methods can be divided into two categories based on the parameter estimation strategy: (1) regression-based and (2) distance-based (He et al., 2011). Regression-based methods derive the relationship between basin characteristics and model parameters through fitted regression models. These mathematically defined relationships are further applied to estimate model parameters of ungauged basins (e.g., Kaspar, 2004; Müller Schmied et al., 2021). A significant drawback of regression-based regionalization is the difficulty of incorporating parameter interdependencies (Poissant et al., 2017), as regression-based approaches often assume that the dependent variables, i.e., the model parameters, are not correlated (Wagener et al., 2004). Distance-based approaches transfer complete parameter sets from similar or nearby donor basins to ungauged basins (e.g., Beck et al., 2016; Nijssen et al., 2000; Widén-Nilsson et al., 2007). Using an ensemble of donor basins, e.g., by averaging the parameter values or model outputs, can improve the performance of such methods (e.g., Arsenault & Brissette, 2014). A significant disadvantage of such methods is the clustering problem of ungauged basins, i.e., the unequal distribution of gauging stations worldwide (Krabbenhoft et al., 2022). Thus, basins exist where distance-based approaches will use incomparable basins to transfer parameter values due to the lack of close basins.

Recent advances have implemented machine learning-based techniques in the context of regionalization. For example, Chaney et al. (2016) used regression trees as an alternative to least squares regression to estimate parameter values in ungauged basins. Pagliero et al. (2019) explored supervised and unsupervised clustering methods to define the similarity of basins to transfer parameter sets. To the authors' knowledge, no study has compared several traditional regionalization methods with machine learning-based methods for a GHM on a global scale.

Some regionalization methods do not make a clear distinction between calibration and regionalization. For exam-
ple, Arheimer et al. (2020) applied a basin grouping beforehand. Then, they jointly calibrated the group members
to define representative parameter sets. Subsequently, the representative parameter sets are transferred to other
basins based on grouping rules. Another approach defines so-called transfer functions (Samaniego et al., 2010)
and calibrates meta-parameters instead of the model parameter values (Beck et al., 2020; Feigl et al., 2022). These
methods, where regionalization is part of the calibration process, often require a change in the calibration process
itself, which is challenging for GHMs (Schweppe et al., 2022), for example, due to a lack of code flexibility (e.g.,
Cuntz et al., 2016).
This study proposes an improved regionalization method for the state-of-the-art GHM WaterGAP3 (Eisner, 2016).
It compares traditional regionalization methods with machine learning-based methods and uses a benchmark-to-
beat and an ensemble of split-sample tests to evaluate the applied methods. Further, global runoff simulations are
compared to analyze the impact of regionalization methods. The overall research topic is evaluating and selecting
regionalization methods for a GHM. Specifically, the study has two objectives. It aims
(1) to propose an improved regionalization method for WaterGAP3 and
(2) to evaluate the impact of regionalization methods on global runoff simulations.
**2. Data and Methods**
**2.1 The Model: WaterGAP3**
The GHM WaterGAP3 simulates the terrestrial water cycle, including the main water storage components and a
simple storage-based routing algorithm. It is a fully distributed model that operates on a five arcmin grid and
simulates at a daily time step. A more detailed description of the model can be found in Eisner (2016).
In WaterGAP3, most model parameter values are set a priori, e.g., using look-up tables for albedo or rooting depth.
Only one parameter, γ, is calibrated, which is part of the soil moisture storage in which runoff generation processes
are present. The model equation for γ, which originates from the HBV-96 model (Lindström et al., 1997), is given
in Eq. (1) (cf. ll. 1223-4 in daily.cpp of the published model (Flörke et al., 2024)). Generally, higher values of γ
lead to lower runoff volumes, while lower values of γ lead to higher runoff volumes. The model parameter is
calibrated per basin within the range of 0.1 and 5. The objective function of the calibration is to minimize the
deviation between the mean annual simulated and observed river discharge, i.e., the calibration aims to reduce the
error in discharge volume. Given the monotonic relationship between the model's parameter and the optimization
function, a simple search algorithm is applied: The parameter space is divided into rectangles, which are subse-
quently subdivided into smaller rectangles depending on the direction γ should be modified to achieve closer
alignment with the optimization target. The calibration results in one calibrated γ value between 0.1 and 5 per
basin. After the calibration, a correction is applied to account for high errors in the mass balance, e.g., due to
inaccuracies in global meteorological forcing products. This correction is only applicable on gauged basins. It is,
therefore, neglected in this study.
$$R = P_t \cdot \left(\frac{S_s}{S_{s,max}}\right)^\gamma \qquad\qquad (1)$$
where $R$ is the daily runoff, $P_t$ is the daily throughfall, $S_s$ is the actual soil storage, $S_{s,max}$ is the maximal soil
storage (given as a global map in Appendix A), and $\gamma$ is the calibration parameter.

Traditionally, the regionalization process in WaterGAP3 is a simple multiple linear regression (MLR) approach to estimate the calibration parameter γ for ungauged basins (e.g., Döll et al., 2003; Kaspar, 2004). The drawback of MLR regarding parameter interaction can be neglected: As there is only one parameter to estimate, parameter interference does not exist. Instead, the approach offers the advantage of a lightweight, transparent application that can be quickly revised and adapted.

## 2.2 Model Data

WaterGAP3 requires various input data, such as soil information, topography, or information on open freshwater bodies. This study uses the same input data as Kupzig et al. (2023). For meteorological forcing, we use the global data set EWEMBI (Lange, 2019). This data product includes daily global forcing data with a spatial resolution of 0.5 degrees (latitude and longitude) that covers a period from 1979 to 2016. Specifically, WaterGAP3 uses the following forcing information from the EWEMBI data set as input:

- daily mean temperature,
- daily precipitation,
- daily shortwave downward radiation, and
- daily longwave downward radiation.

The WaterGAP3 calibration requires observed monthly river discharge data. This discharge data is subsequently transformed into annual discharge sums and used as a benchmark in the calibration procedure. In this study, we used discharge data from 1,861 stations that were manually verified (Eisner, 2016). To get the best data available, we have updated all available station data with recent data from The Global Runoff Data Center (GRDC, 2020). All stations have at least five years of complete (monthly) station data between 1979 and 2016. For each station, a contribution area, i.e., a basin, is defined with the gridded flow-direction information obtained from WaterGAP3, based on the HydroSHEDS database (Lehner et al., 2008).

The 1,861 basins are calibrated using the above-described standard calibration approach for WaterGAP3. Following the standard calibration procedure, some basins still have an insufficient model performance. In this context, we define a monthly Kling-Gupta-Efficiency (KGE) (Gupta et al., 2009) below 0.4 or more than 20 % bias in monthly flow as insufficient model performance. The expression for the KGE is given in Eq. (2). We underscore the importance of minimizing the error in discharge volume by defining it as an additional criterion corresponding to the optimization target during calibration. Basins not fulfilling the defined conditions regarding bias and KGE are neglected in further analysis to avoid high parameter uncertainty due to errors in input data, model structure, or discharge data affecting the analysis. Further, we have excluded all basins with less than 5000 km$^2$ (inter-) basin size from the next upstream basin. We assume that this inter-basin size is large enough to assume a certain degree of interdependency between nested basins. In total, 933 out of 1,861 basins are selected for regionalization (626 are neglected due to insufficient model performance, and 302 are neglected due to inadequate basin size).

$$KGE = 1 - \sqrt{(1-r)^2 + \left(1 - \frac{\sigma_y}{\sigma_x}\right)^2 + \left(1 - \frac{\mu_y}{\mu_x}\right)^2} \qquad (2)$$

where $r$ is the Pearson correlation coefficient between observed discharge x and simulated discharge y, $\sigma$ denotes the corresponding standard deviation, and $\mu$ the corresponding mean of observed and simulated discharge.

Figure 1a depicts the worldwide calibrated basins, highlighting gauged and ungauged regions. Whereas most parts
of North and South America are gauged, Africa and Australia remain largely ungauged. A cluster of gauged basins
is in Central Europe and in Eastern Asia. Gauged regions with insufficient model performance are mainly in the
Mississippi River basin, Southern Africa, Australia, and large parts of Brazil. These regions are known to be chal-
lenging for GHMs (e.g., cf. Fig. 8b in Stacke & Hagemann, 2021).
Figure 1b shows the calibrated values for γ. It emerges that the calibrated values tend to be at the upper and lower
bounds of the parameter space. This behavior is already known (cf. Fig. 4b in Müller Schmied et al., 2021). A
brief sensitivity analysis and discussion of the calibration parameter are included in Appendix B. The results of
this analysis indicate that the clustering of the calibrated parameter value is not related to an inappropriate selection
of the parameter bounds but instead to the absence or an insufficient representation of processes. Thus, the clus-
tering of the calibrated values does not indicate an inadequate selection of the parameter bounds but highlights the
necessity to improve the model structure and the calibration strategy for WaterGAP3. However, this study focuses
solely on analyzing and implementing regionalization methods. It does not aim to enhance the model structure or
to change the calibration procedure of WaterGAP3. Future studies are needed to achieve the latter, as WaterGAP3
contains many hard-coded parameters or parameters defined by look-up tables that need to be analyzed to identify
and adjust sensitive parameters more accurately during calibration. Initial steps in this direction have already been
taken for WaterGAP2 in the form of a multivariate and multi-objective case study in the Mississippi River basin
(Döll et al., 2024).

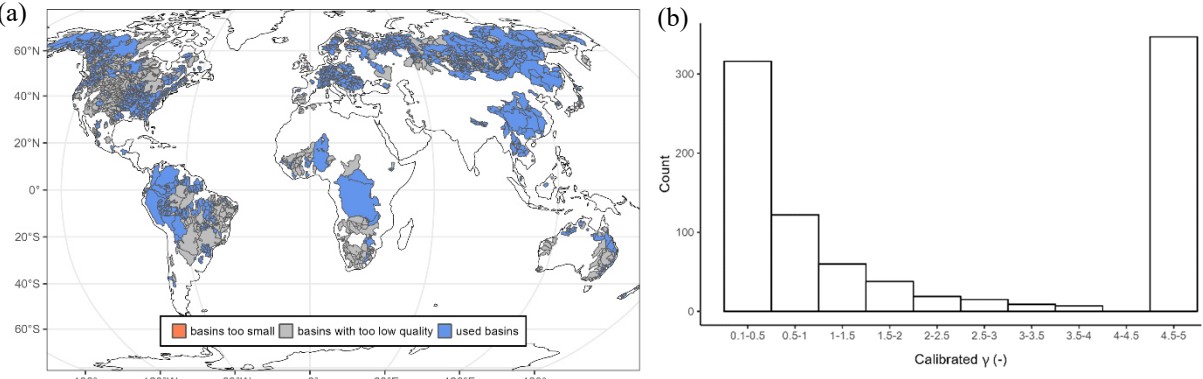

**Figure 1: a) Map of calibrated basins, highlighting basins not used for regionalization due to insufficient model perfor-**
**mance or inadequate basin size and b) the histogram of the calibrated γ values for all used basins showing a cluster of**
**parameter values at the parameter bounds.**
**2.3 Basin Descriptors**
This study uses basin descriptors as predictors to drive regression-based or distance-based regionalization ap-
proaches. These basin descriptors are based on data used within the model simulation (as they are globally avail-
able). They are aggregated to basin values using a simple mean method to have the same spatial resolution as the
calibrated model parameter. Thus, in the case of nested basins, the inter-basin area is used to define the basin
descriptors. The selection of the predictors, i.e., basin descriptors that support the estimation of γ, is crucial for
regionalization methods (Arsenault & Brissette, 2014). Typically, this selection aims to obtain the most infor-
mation with the least number of predictors to (1) improve the model quality and (2) limit over-parametrization. In
this study, we use 12 basin descriptors to develop regionalization methods; nine of these descriptors are physio-
graphic, while the remaining three are climatic (see Table 1). Most descriptors are not correlated (see Appendix
C), i.e., we minimize redundant information (Wagener et al., 2004).

A descriptor subset is selected based on correlation analysis between basin descriptors and calibrated γ value and entropy assessment. Pearson's correlation coefficient detects linear correlation, and Spearman's Rho and Kendall's Tau detect a non-linear correlation. Shannon entropy (Shannon, 1948) measures the information gain of the predictors explaining the calibrated γ value. The higher the information gain, the more valuable the basin descriptor is for explaining the variation in the calibrated γ value. The analysis directly evaluates the relationship between the calibrated parameter and the basin descriptors, as WaterGAP3 uses only one calibration parameter with a clear global optimum within the parameter space. An alternative would be to use flow characteristics to define the basis for regionalization (e.g., Pagliero et al., 2019). We decided to use the calibrated parameter instead of flow characteristics as it does not need any further assumption on which flow characteristics determine the model's parameter.

Statistical information of the evaluated basin descriptors and the corresponding correlation coefficients and information gain are listed in Table 1. The basin descriptors demonstrate a considerable degree of variability, e.g., the basin size ranges from 5000 km$^2$ to 3,112,480 km$^2$ with a median of 13,796 km$^2$. The mean temperature varies from -19 °C to 29 °C, and the sum of precipitation ranges from 213 mm to 5,716 mm. Although there is a high degree of variability in the analyzed basin descriptors, the basin descriptors exhibit low correlation coefficients with the calibrated values. For example, the permafrost coverage shows the strongest Pearson correlation of -0.37 (and -0.50 for Spearman's Rho). The information gain indicates the same results as the correlation analysis, i.e., the information gain is generally relatively low, and descriptors with a higher correlation tend to have a higher information gain. For example, the mean temperature exhibits the maximal information gain of 17.6 % and has the second-highest correlation coefficient with a Pearson correlation of 0.34.

**Table 1: Basin descriptors: statistical information, correlation, and entropy assessment. Selected physiographic and climatic basin descriptors are written in bold.**

| | Basin Descriptor | Attribute Information | | | | Entropy & Correlation | | | |
|---|---|---|---|---|---|---|---|---|---|
| | | Min | Max | Mean | Median | IG (%)[1] | Pearson | Spearman | Kendall |
| physiographic | Soil Storage (mm) | 12.405 | 610.469 | 220.805 | 195.778 | 13.07 | -0.21 | -0.15 | -0.11 |
| | Open Water Bodies (%) | 0.000 | 63.960 | 5.521 | 1.812 | 5.65 | -0.01 | -0.08 | -0.05 |
| | Wetlands (%) | 0.000 | 63.466 | 4.164 | 0.547 | 5.01 | -0.02 | -0.13 | -0.09 |
| | Size (km$^2$) | 5000 | 3,112,480 | 37,572 | 13,796 | 1.42 | -0.04 | -0.04 | -0.03 |
| | **Slope Class (-)** | 10.057 | 67.756 | 38.668 | 38.364 | 16.60 | -0.31 | -0.37 | -0.27 |
| | Altitude (m.a.s.l.) | 30.239 | 4765.166 | 591.024 | 394.870 | 9.30 | -0.18 | -0.28 | -0.20 |
| | Sealed Area (%) | 0.000 | 12.3 | 0.6 | 0.1 | 4.49 | 0.22 | 0.38 | 0.29 |
| | **Forest (%)** | 0.000 | 100.000 | 35.340 | 24.002 | 13.82 | -0.25 | -0.18 | -0.14 |
| | **Permafrost & Glacier (%)** | 0.000 | 95.000 | 16.662 | 0.000 | 13.12 | -0.37 | -0.50 | -0.40 |
| climate | **Mean Temperature(°C)** | -18.848 | 28.823 | 7.720 | 7.707 | 17.56 | 0.34 | 0.41 | 0.30 |
| | Yearly Precipitation (mm) | 213.6 | 5,716.3 | 996.5 | 779.5 | 9.23 | 0.02 | 0.21 | 0.14 |
| | **Yearly Shortwave Downward Radiation (Wm$^{-2}$)** | 1,050.6 | 3,043.2 | 1,857.9 | 1,759.7 | 15.79 | 0.31 | 0.33 | 0.24 |

[1]Information gain is given in percentage of total information content in γ after Shannon (1948)

In contrast to the findings of Wagener and Wheater (2006), the correlation coefficients between the basin descriptors and the calibrated values are relatively low, indicating a weak relationship. One potential explanation for this discrepancy is that Wagener and Wheater (2006) used a smaller number of basins in southeast England, with limited versatility (e.g., regarding climate and seasonality) compared to the 933 worldwide basins used in this study. Studies using a large number of basins likely tend to find a lower correlation between catchment attributes and model parameters (Merz et al., 2004). Moreover, the clustered calibrated γ values at the bounds of the valid parameter space may disturb the results of this analysis. As the calibrated value masks the effect of multiple sources

of errors, such as uncertainty in the input data, model structure, or varying hydrological processes, finding a mean-
ingful relationship between catchment characteristics and calibrated values is challenging.
Because the basis for the descriptor selection seems uncertain, given the low correlation and the named constraints,
we additionally run the regionalization methods with all descriptors to evaluate the descriptor selection. Further
on, to ascertain the advantage of integrating climatic descriptors, we run the regionalization methods using either
physiographic or climatic descriptors. In total, we used four groups of basin descriptors to implement the region-
alization methods:
- "cl": all three climatic descriptors,
- "p": all nine physiographic descriptors,
- "p+cl": all 12 descriptors, and
- "subset": two correlated climatic descriptors (mean temperature, annual shortwave radiation) & three
correlated physiographic descriptors (slope class, forest %, permafrost %).

**2.4 Regionalization Methods**

In our study, we test several traditional and machine learning-based regionalization methods against each other
and a defined benchmark-to-beat to find suitable regionalization methods for WaterGAP3. At the global scale,
regionalization is particularly challenging due to (1) the lack of high-quality data, (2) the diversity of dominant
hydrological processes in basins, and (3) the high computational demands of the models. Therefore, a robust re-
gionalization method that applies to a wide variety of basins and is not computationally demanding should be
selected for a global application.
We test three common traditional approaches and two machine learning-based approaches using the concepts of
spatial proximity, physical similarity, and regression-based methods. As WaterGAP3's model calibration is very
rigid and has only one parameter, it is not feasible to implement and test regionalization methods that incorporate
regionalization into the calibration process, such as transfer functions. In addition, we avoid high computational
demands as all evaluated methods are applicable after the calibration, i.e., without running the model.
As the calibration of WaterGAP3 results in a parameter distribution with a cluster of parameter values at the
parameter bounds, we implement a so-called "tuning" to introduce information about the parameter space into
regionalization. In detail, we apply a simple threshold-based approach to shift the regionalized parameter values
to the extremes, i.e., $\gamma_{est} < \gamma_1 \rightarrow \gamma_{reg} = 0.1$ and $\gamma_{est} > \gamma_2 \rightarrow \gamma_{reg} = 5.0$. The thresholds $\gamma_1$ and $\gamma_2$ are defined
by applying the k-means algorithm with three centers to the calibrated parameter values. This clustering results in
three clusters: one for low, one for medium, and one for high $\gamma$ values. Subsequently, $\gamma_1$ refers to the highest $\gamma$
value of the low cluster and $\gamma_2$ refers to the lowest $\gamma$ value of a high cluster.
To evaluate the regionalization methods, we implement an ensemble of split-sample tests. Specifically, we ran-
domly split the basins into 50 % gauged (for training) and 50 % pseudo-ungauged (for testing). The split has a
relatively high percentage of pseudo-ungauged basins, accounting for many missing gauges worldwide and the
high importance of generalizability. We fit the methods and apply them to the training and testing data sets. The
split-sample test is repeated 100 times by randomly splitting the basins to account for sampling effects.
As there is only one calibration parameter, $\gamma$, this parameter has a global optimum per basin. Consequently, the
quality of training and testing is directly assessed by the deviation between the regionalized and the calibrated
value for γ. The closer the regionalized values are to the calibrated ones, the more accurate the prediction. We
assess the prediction accuracy by the logarithmic version of the mean absolute error (logMAE) shown in Eq. (3)
to account for the decreasing sensitivity of γ for higher values (see Appendix B). The lower the logMAE, the better
the prediction; a zero value in logMAE expresses no error. The regionalization method is robust if the prediction
accuracy is similar in training and testing. A generally good performance, i.e., small logMAE values, indicates
that the regionalization method suits WaterGAP3. The comparison of γ values enables applying a wide range of
regionalization methods and sets of descriptors, as no computationally intensive model simulation is required.
However, it assumes that deviations in γ lead, in turn, to deviations in discharge, which is only partially true
because of varying parameter sensitivity in basins (e.g., Kupzig et al., 2023). To validate that the logMAE is a
sufficient approximator for the regionalization performance in WaterGAP3, we use one representative split-sample
from the ensemble to compare the accuracies in simulated discharge for different regionalization methods.
$$logMAE = \frac{1}{n}\sum\left|\ln\left(\gamma_{x,i} + 1\right) - \ln(\gamma_{y,i} + 1)\right| \qquad (3)$$
where $n$ is the number of basins in the corresponding sample, $\gamma_{x,i}$ is the calibrated value of γ for the i[th] basin, and
$\gamma_{y,i}$ is the estimated value of γ for the i[th] basin. We applied a Box-Cox-type transformation with $\lambda_1=0$ and $\lambda_2=1$
(Box and Cox, 1964) to calculate the logMAE, avoiding negatively transformed values.

**Regression-based methods**

The traditionally used regionalization approach of WaterGAP3 is a regression-based MLR. As the benchmark-to-
beat, we use the regionalization approach from WaterGAP2.2d defined in Müller Schmied et al. (2021). We con-
sider it a suitable benchmark-to-beat given that WaterGAP2 has a model structure and calibration process that is
very similar to WaterGAP3. The main difference between these models is that WaterGAP2 simulates at 0.5°spatial
resolution. The benchmark-to-beat consists of "a multiple linear regression approach that relates the natural loga-
rithm of γ to basin descriptors (mean annual temperature, mean available soil water capacity, fraction of local and
global lakes and wetlands, mean basin land surface slope, fraction of permanent snow and ice, aquifer-related
groundwater recharge factor)". (Müller Schmied et al., 2021) We fit this regression model to our data and define
the quality of this approach as the benchmark-to-beat. Moreover, we test an independent MLR approach without
using the logarithmical scaling of γ and using the above-defined sets of basin descriptors. For MLR and the bench-
mark-to-beat, we use the lm() function of the R package stats (R Core Team, 2020). After applying the regression
model, we adjust the estimated parameter values to ensure that the estimated values range between 0.1 and 5.
Furthermore, a machine learning-based method, random forest (RF), is tested for regionalization as an alternative
to MLR. Here, we implement the random forest algorithm with the randomForest() function from the R package
randomForest (Liam & Wiener, 2002), which is based on Breimann (2001). The algorithm uses an ensemble of
decision trees, making the decision human-like. It is relatively robust because it incorporates random effects into
the training process. To implement this randomness, we define the algorithm as one that can choose between two
randomly selected predictors at each node, using an ensemble of 200 trees.

**Physical Similarity**

As the traditional physical similarity approach, we use Similarity Indices (in the following named with SI), apply-
ing the methodology proposed by Beck et al. (2016). The SI (see Eq. (4)) are derived using the defined basin
descriptors sets, and the parameter of the most similar basin is transferred to the pseudo-ungauged basin. Addi-
tionally, we use an ensemble of basins to control whether an ensemble-based approach leads to more robust results.
The optimal number of donor basins may vary between research regions and hydrological models (Guo et al.,
2020). Here, we use ten donor catchments (noted with "ensemble") based on Beck et al. (2016) and McIntyre et
al. (2005). Further, we apply a simple mean method for the ensemble-based prediction to aggregate the ensemble
of $\gamma$ values into one predicted parameter value.
$$S_{i,j} = \sum_{p=1}^{n} \frac{|Z_{p,i} - Z_{p,j}|}{IQR_p} \qquad (4)$$
where $S_{i,j}$ is the Similarity Index between basin $i$ and basin $j$, $Z_{p,j}$ is the basin descriptor $p$ for basin $j$, $IQR_p$ is the
interquartile range for basin descriptor $p$ among all (gauged) basins, and $n$ is the number of all basin descriptors
used.
As an alternative machine learning-based approach, we apply a simple k-means algorithm. We selected the k-
means algorithm because it is one of the most widely used clustering algorithms (Tongal & Sivakumar, 2017). It
is easy to understand and use. The algorithm kmeans() is implemented in the R base package stats. It aims to
maximize variation between groups and minimize variation within groups. The number of clusters to use is deter-
mined by multiple indices calculated with the R package NbClust (Charrad et al., 2014). For all 933 basins and
the defined sets of basin descriptors, most indices defined three as the optimal number of clusters. Accordingly,
we use three clusters to generate the groups of basins. As different scales of the predictor values can affect the
clustering, a rescaling with min-max-normalization (see Eq. (5)) is performed on the training set and applied to
the testing set. After the grouping, the mean $\gamma$ value is assigned as a representative calibrated value to the corre-
sponding basin group. To estimate the corresponding group for a pseudo-ungauged basin, the knn algorithm is
used, and the representative $\gamma$ value of the group is assigned to the pseudo-ungauged basin. This algorithm is
implemented by the knn() function of the R package class (Venables & Ripley, 2002). Since the k-means method
is less flexible than SI, we implement a highly flexible version, using the knn algorithm directly to define the donor
basin most similar to each ungauged basin. Using the knn algorithm directly, we test how beneficial it is to create
groups of similar basins using the kmeans algorithm and regionalize the parameter with a representative mean
value.
$$Z'_{p,j} = \frac{Z_{p,j} - \min_{j \to m}(Z_{p,j})}{\max_{j \to m}(Z_{p,j}) - \min_{j \to m}(Z_{p,j})} \qquad (5)$$
where $Z'_{p,j}$ is the normalized basin descriptor $p$ for basin $j$, $Z_{p,j}$ is the basin descriptor $p$ for the basin $j$, $m$ is the
number of (gauged) basins.
**Spatial Proximity**
The spatial proximity approach is one of the easiest to regionalize parameter values. However, it is also often
criticized that nearby basins do not necessarily have the same hydrological behavior (Wagener et al., 2004). Fur-
thermore, its performance depends on the density of the network of gauged basins (Lebecherel et al., 2016). The
dependency on network density is particularly challenging for global applications where large parts of the world
are ungauged (e.g., northern Africa). Nevertheless, the approach has been successfully applied in other studies
(e.g., Oudin et al., 2008; Qi et al., 2020), even globally (Widén-Nilsson et al., 2007). Here, we take the distance
between the centroids of the basins as the reference for the spatial distance between basins, as done by others
(Oudin et al., 2008; Merz and Blöschl, 2004). We use the abbreviation SP in the text below to refer to the spatial
proximity approach. Figure 2 provides an overview of the applied regionalization methods and information used
for the experimental setup.

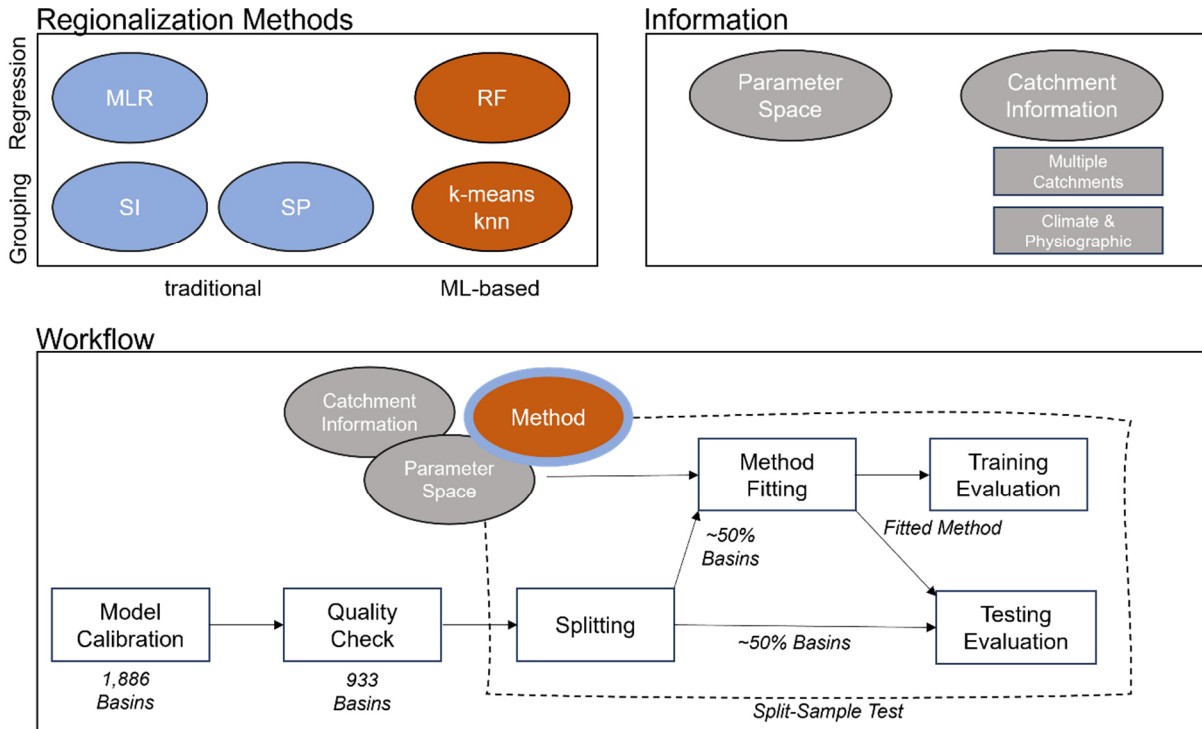

**Figure 2: Experimental setup of the study: regionalization methods, used modifications and information, and the general workflow (MLR: Multiple Linear Regression, SI: Similarity Indices, SP: Spatial Proximity, RF: RandomForest).**

## 3. Results and Discussion

### 3.1 Evaluating the effect of tuning

First, the impact of the tuning approach on the regionalization approaches is evaluated. Therefore, Fig. 3 depicts
the differences in logMAE between the standard and tuned approaches in testing, i.e., using the pseudo-ungauged
basins. A positive difference in logMAE indicates an increase in accuracy, whereas a negative difference indicates
a decrease in accuracy due to the tuning.
Using the tuning thresholds of about 1.1 and 3.4 for $\gamma_1$ and $\gamma_2$, respectively, enhances the predictive accuracy for
kmeans, MLR, RF, and the ensemble approach of SI. The most remarkable improvement for kmeans, RF, and SI
ensemble is achieved when all physiographic descriptors are used as input (mean improvement of 0.077, 0.058,
and 0.071, respectively). MLR shows the most significant improvement when using all available descriptors (mean
improvement of 0.038). In contrast, the tuning decreases the performance for knn, SI, and SP, with a mean degra-
dation between -0.02 and -0.05. Unlike the enhanced regionalization techniques, these methods transfer single-
basin information to ungauged regions. Thus, the tuning disturbs the use of single-basin information yet simulta-
neously enhances the performance of methods that transfer multi-basin information. The disturbance or improve-
ment is probably related to the capability of the methods representing the clustering of parameter values at the
extremes: Whereas the multi-basin information transfer implies a smoothing and thus suffers from a lack of rep-
resenting the extremes, the single-basin information transfer exhibits no such a smoothing.

The exception from the above-defined rule is the benchmark-to-beat approach. The benchmark-to-beat is the only approach that uses logarithmic scaled γ values when fitting the model. This logarithmic transformation leads to an increase in estimating small values. Thus, when the benchmark-to-beat is tuned, more basins with higher calibrated γ values receive low estimates. The tuning intensifies this effect, leading to a decrease in the accuracy of the logMAE from the standard to the tuned version. Thus, for models using logarithmical transformed γ values, the defined thresholds for the tuning are not appropriate.

Applying knowledge of the optimal parameter space enhances the quality of regionalization for methods transferring multi-basin information in case the tuning thresholds are appropriate. This positive effect is not surprising, as incorporating a priori information about parameter distribution strengthens parameter estimation (e.g., described in Tang et al. (2016) using the Bayes Theorem). However, for single-basin transfer, which already represents the parameter space well, i.e., the clustering of γ at the extremes, the tuning disturbs the performance. This indicates that such tuning needs to be cautiously introduced as there is the risk of decreasing the accuracy of regionalization.

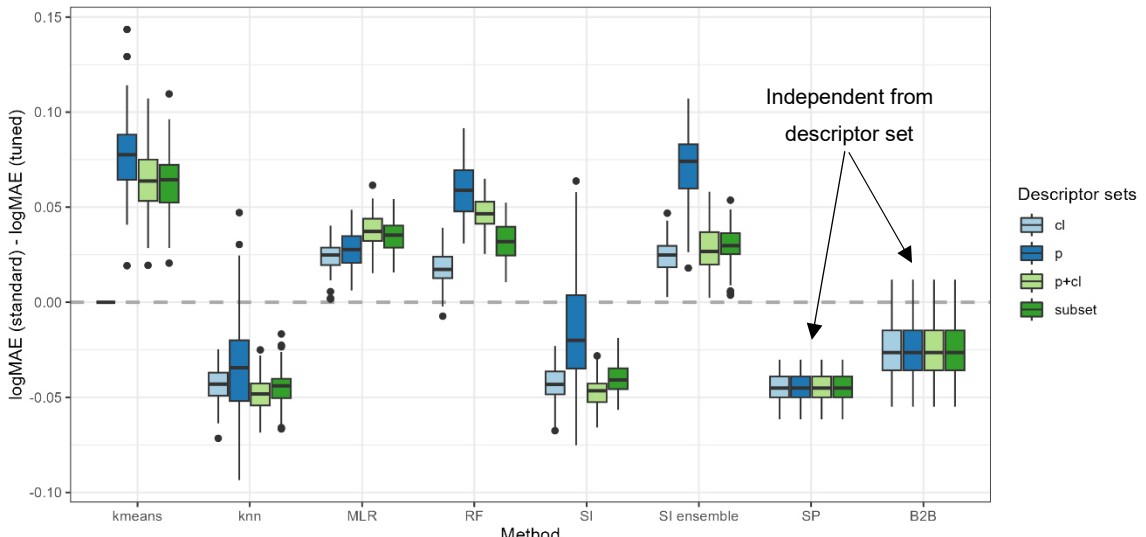

**Figure 3: Changes in performance between standard and tuned versions for all applied regionalization approaches. Positive values indicate an improvement related to the tuning.**

### 3.2 Evaluating descriptor subsets & algorithm selection

Different descriptor sets yield different performances in regionalizing γ. Table 2 shows the median of all logMAE values for the testing. For a complete overview of the results of the split-sample test ensemble, see Appendix D. Evaluating Table 2 reveals that the selected subset or all descriptors consistently yield the best performance across all regionalization methods. In both variants of the ensemble approach of SI, the tuned version of the no-ensemble approach of SI, and the standard version of RF, the selected subset yields the best results. For all other methods, using all descriptors yields the best results. Hence, all methods perform best when combining climatic and physiographic descriptors. This benefit of using climatic and physiographic descriptors is consistent with others that often apply a combination of climatic and physiographic descriptors, achieving optimal regionalization results (e.g., Oudin et al., 2008; Reichl et al., 2009).

The machine learning-based approaches seem to benefit most when using more information displaying an improvement for all methods (knn, kmeans, and RF) and both variants (standard and tuned) ranging from "cl", "p", "subset" to "p+cl". This is not surprising as machine learning is developed to deal with big data sets. The traditional

methods MLR and SI do not exhibit such a distinct pattern. The (weakly) correlated subset of climatic and physiographic descriptors yields the best results for SI. As utilizing all descriptors decreases the performance slightly, the results indicate that uncorrelated descriptors may disturb the performance of this approach. For MLR, the meaning of physiographic information is highest, resulting in the best ("p+cl") and second best ("p") results. The disparate performance of the regionalization methods when using different descriptor sets indicates that different methods use descriptor sets with varying efficiency. It also emphasizes that the selection of descriptors impacts the regionalization method's results, as noted by others (Arsenault & Brissette, 2014). Consequently, the above-performed analysis defining a descriptor subset lacks universal validity as methods exist where the defined subset is outperformed. Instead, the validity of this approach is most closely aligned with the SI approaches.

Although the algorithms kmeans and knn are similar, they yield considerably different performances in Table 2. As knn shows a logMAE of 0.432 at best, the kmeans algorithm performs poorly, resulting in the best logMAE of 0.472. This indicates that applying the kmeans clustering algorithm to transfer averaged parameters is inappropriate for WaterGAP3. This may be attributed to the reduced flexibility of the approach, which entails estimating only three $\gamma$ values due to the optimal, though limited, number of centers. The ensemble SI approach consistently outperforms the no-ensemble SI approach in almost all variants. The positive effect of an ensemble approach for SI has already been noted (Oudin et al., 2008). Therefore, it is recommended that the number of donor basins derived from the literature be adopted in future applications to be optimal for WaterGAP3, likely resulting in higher performance.

**Table 2: Median logMAE of 100 split-samples for pseudo-ungauged basins, i.e., in testing, for all regionalization methods applying four sets of descriptors for a) the standard version and b) the tuned version. The bold numbers indicate a better performance than the benchmark-to-beat. Thicker edges mark best-performing variants, which are chosen for further analysis. Grey-shaded cells indicate worst-performing variants, which were taken to validate the assumption that lower logMAE values result in lower KGE values.**

(a)

| test (median) | MLR | RF | SI | | kmeans | knn | SP | B2B |
|---|---|---|---|---|---|---|---|---|
| | | | no ens. | ensemble | | | | |
| cl | 0.552 | 0.483 | 0.496 | 0.483 | 0.619 | 0.501 | | |
| p | 0.479 | 0.465 | 0.487 | 0.480 | 0.551 | 0.477 | **0.454** | 0.461 |
| p+cl | 0.464 | 0.464 | **0.454** | 0.462 | 0.534 | **0.432** | | |
| subset | 0.488 | 0.488 | 0.461 | **0.439** | 0.539 | 0.467 | | |

(b)

| test* (median) | MLR | RF | SI | | kmeans | knn | SP | B2B |
|---|---|---|---|---|---|---|---|---|
| | | | no ens. | ensemble | | | | |
| cl | 0.529 | **0.467** | 0.537 | **0.459** | 0.619 | 0.546 | | |
| p | **0.441** | **0.416** | 0.532 | **0.455** | 0.515 | 0.521 | 0.502 | 0.488 |
| p+cl | **0.427** | **0.403** | 0.503 | **0.435** | 0.472 | 0.480 | | |
| subset | **0.453** | **0.408** | 0.501 | **0.409** | 0.477 | 0.509 | | |

Only a few regionalization methods outperform the benchmark-to-beat. The best descriptor sets of tuned MLR, RF, and SI ensemble approach have a logMAE of 0.427, 0.403, and 0.409, respectively. The standard version of knn ("p+cl") and SP yield 0.432 and 0.454 in logMAE, respectively. Additionally, two variants of the standard SI approaches outperform the benchmark-to-beat yet exhibit inferior results compared to the selected tuned approach.

All other regionalization methods show higher logMAE values than the benchmark-to-beat. These methods are considered insufficient in terms of performance to regionalize γ in WaterGAP3. As the benchmark-to-beat outperforms all kmeans approach variants, it is deemed unsuitable for regionalizing γ for WaterGAP3 and, therefore, excluded from further analysis.

The well-performing SP on a global scale is surprising as the distances between basins are potentially long, and hydrological processes may strongly vary. It is probably beneficial for the SP approach that γ comprises all kinds of errors, e.g., spatially localized errors in global forcing products (e.g., Beck et al., 2017 reported errors for arid regions in the precipitation product) or inaccurately represented processes for larger regions. Thus, the estimation of γ might be appropriate, but not because of the same hydrological behavior but due to the same kind of errors.

The RF approach is outstanding, as it shows a massive loss in performance from training to testing (see Appendix D). In detail, the logMAE in testing is about twice the logMAE in training. In comparison, other methods show values of logMAE in testing ranging from 95.6 % to 101.4 % of logMAE in training. This performance loss indicates that RF is not a robust regionalization method for WaterGAP3. Other studies that reported the good performance of RF for regionalization have not investigated the stability of the performance from training to testing (Golian et al., 2021; Wu et al., 2023). Likely, the mathematical problem of predicting the calibrated parameter for WaterGAP3, with all its challenges (e.g., tailored parameter space, clustered calibrated parameter, and incorporation of many sources of errors), cannot be adequately solved by RF. Thus, although RF is known to be especially robust among other machine learning-based techniques, it shows symptoms of over-parameterization. This indicates that the algorithm is too flexible and adjusts to noise in the data, missing the underlying systematic. This lack of robustness is particularly disadvantageous since, for WaterGAP3, regionalization is applied globally, requiring regionalizing large parts of the world. In consequence, the RF approach is left out from further analysis and defined as not suitable to regionalize γ for WaterGAP3.

For the tuned MLR approach and the knn approach, the best performing and, therefore, selected variant employs all 12 descriptors. This number of predictors for a regionalization method is among the highest found in the literature (e.g., McIntyre et al., 2013, used three predictors; Beck et al., 2016, used eight predictors; Chaney et al., 2010, used 13 predictors). In general, it is advisable to limit the number of degrees of freedom in a model to reduce the risk of over-parametrization, thus increasing the probability of generalizability (Seibert et al., 2019). As both model variants exhibit a stable model performance during training and testing (see Table D1), using a high proportion of the basins for testing, i.e., 50 %, we consider the two variants robust despite the relatively high number of predictors used. Therefore, we consider them appropriate for further model evaluation.

Nevertheless, the chosen basin descriptors for knn and tuned MLR could be enhanced in future studies. As the descriptor set "p+cl" was initially considered as a control group to determine the suitability of the selected subset, it is not optimal. To indicate potential enhancements regarding the descriptor set for both methods, we calculated a simple permutation-based feature importance score (cf. Breiman, 2001) by randomly shuffling each predictor within the testing data set and quantifying the loss in logMAE relative to the logMAE of the original testing data set. The higher the loss, the more critical the shuffled predictor for the regionalization method. The resulting feature importance scores are presented in Appendix E, indicating that for the tuned MLR, the subset of (weakly) correlated descriptors should be extended by including waterbody information. For the knn approach, the calculated feature importance scores indicate that it should be extended by including information about the soil storage.

**3.3 Performance of selected algorithm in pseudo-ungauged basins**

To avoid the high risk of sampling effect when applying the split-sample test, we conduct an ensemble of 100 split-sample tests analyzing the median of logMAE between regionalized and calibrated values as an indicator for performance. Directly using the differences in regionalized and calibrated values is only meaningful when the calibrated value represents the global optimum. As this is often not the case, e.g., due to equifinality, the performance of regionalization methods is usually assessed by the accuracy of simulated discharge (e.g., Samaniego et al., 2010; Arsenault & Brissette, 2014). Because WaterGAP3 requires computationally intensive simulations, running WaterGAP3 for all 100 split-sample tests for the selected methods is not feasible. Therefore, we select a single representative split-sample to assess the quality of representing the discharge in the pseudo-ungauged basins using regionalized γ values. The representative split-sample leads to comparable logMAE values to the corresponding median of the ensemble for all regionalization methods. For the evaluation, WaterGAP3 was run for the same period used in calibration (from 1979 to 2016), with the first year simulated ten times to allow for model warm-up. Using this period ensures the availability of sufficient data for the evaluation (see Chapter 2.2). Furthermore, the differences between the monthly simulated and observed discharge are assessed using the KGE.

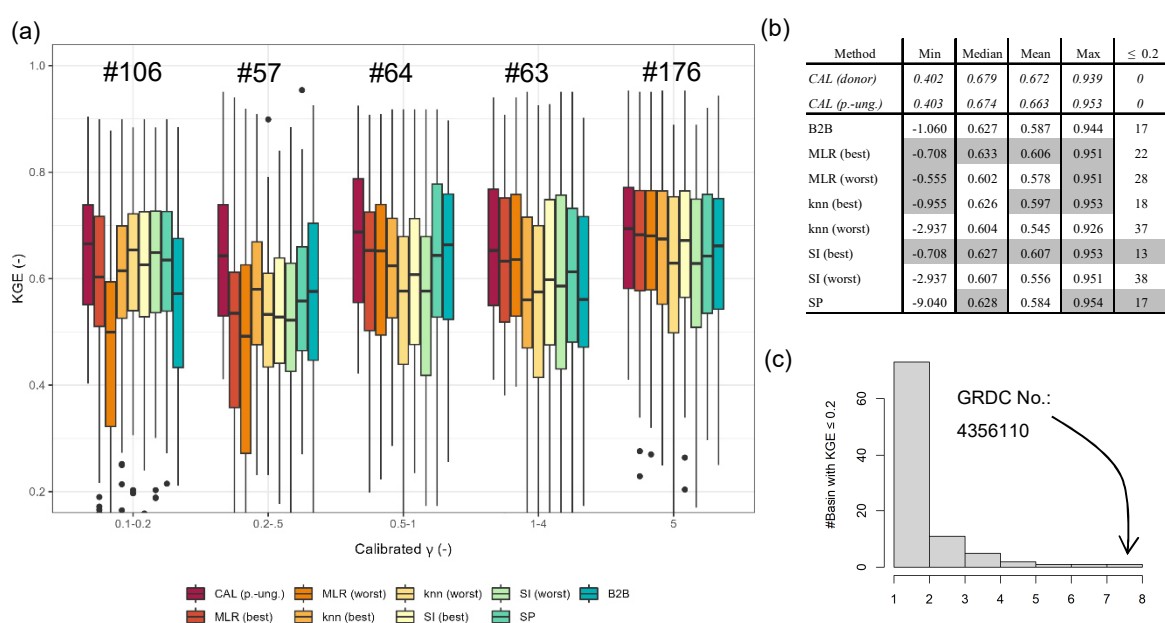

**Figure 4: a) KGE values of pseudo-ungauged basins from split-sample test grouped by the range of calibrated γ values, b) selected metrics of KGE values from the pseudo-ungauged basins (better or equal performance to the benchmark-to-beat is highlighted in grey), and c) histogram of the number of pseudo-ungauged basins with a KGE below 0.2 and the corresponding number of methods exhibiting this performance loss.**

To evaluate the KGE, we select the best-performing methods that outperform the benchmark-to-beat: tuned MLR "p+cl", knn "p+cl", tuned SI ensemble "subset", and SP (see Table 2). For the sake of simplicity, we further mark them with "(best)". Additionally, we select three poorly performing variants to validate the assumption that methods resulting in higher logMAE values tend to result in lower KGE values, i.e., lower accuracy of simulated discharge. These methods are tuned SI "cl" (logMAE: 0.537), tuned knn "cl" (logMAE: 0.546), and MLR "cl" (logMAE: 0.552). Further, we denote these methods with "worst". Applying the selected methods and the benchmark-to-beat method results in eight estimates of γ for the pseudo-ungauged basins, whose performance is further evaluated in terms of simulated discharge accuracy.

Figure 4a shows the resulting KGE values for the evaluated regionalization methods and the calibrated version as
grouped boxplots for different ranges of calibrated γ. The methods show different performances for different γ
ranges, indicating their strengths and weaknesses. For the smallest γ range, "0.1-0.2", the selected methods that
perform well during the split-sample test outperform the benchmark-to-beat. The better result for minimal γ ranges
is probably partially related to the advantage of the tuning, which leads to more predictions of 0.1 within the
regionalization. The benchmark-to-beat shows the best performance for γ values between 0.2 and 0.5. The good
performance for basins with calibrated γ values between 0.2 and 0.5 is probably related to the benefit of using the
logarithmical version of γ in the benchmark-to-beat, leading to more estimates of smaller values. However, this
affects only 12 % of the basins, as calibrated values between 0.2 and 0.5 are not frequently present in the calibration
result. Generally, the differences in KGE appear higher for smaller γ values, probably due to the decreasing pa-
rameter sensitivity with higher values (see Appendix B).
Given the variability in the performance of the regionalization methods across the depicted γ ranges, it is challeng-
ing to identify an overall best regionalization method using Fig. 4a. Therefore, we compare the various metrics of
the KGE values depicted in Fig. 4b. The analyzed metrics are the minimum, maximum, mean, and median. Further,
we count the number of poorly performing basins, defined as basins with a KGE below 0.2. In Fig. 4b, metrics
that exceed the benchmark-to-beat are grey-shaded. Comparing the KGE metrics in Fig. 4b reveals that the meth-
ods showing higher logMAE values in our split-sampling test ensemble also show lower performance in simulating
discharge. For example, all mean (and median) KGE values of the "worst" methods are below the mean KGE of
0.587 from the benchmark-to-beat, ranging from 0.545 to 0.578. This indicates that the used logMAE between
regionalized and calibrated values is a valid tool for a preliminary selection of adequate methods for the regional-
ization of WaterGAP3. However, for a more comprehensive analysis, we recommend additionally analyzing the
accuracy of simulated discharges, as the logMAE of calibrated and regionalized parameter values simplifies the
inherent complexity between model parameters and model performance.
Moreover, SI (best) outperforms the benchmark-to-beat in all listed metrics, reducing poorly performing basins
and enhancing well-performing basins. MLR (best) performs very similarly to SI (best), yet it shows a higher
number of basins with KGE values below 0.2. In comparison to the benchmark-to-beat, it outperforms four out of
five criteria. The remaining well-performing methods, SP and knn (best), demonstrate superior or equal perfor-
mance to the benchmark-to-beat in three out of five criteria. SP results in an equal number of poorly performing
basins, and the minimal KGE value is lower than for the benchmark-to-beat. The knn (best) approach has a slightly
worse median of KGE, i.e., -0.001, and one additional basin shows a KGE below 0.2.
As SI (best) outperforms the benchmark-to-beat in all metrics, we conduct a statistical test to ascertain whether
there is a statistically significant difference in KGE results between the methods. To this end, we use a one-sided
paired Wilcoxon rank sum test to test the null hypothesis of whether the KGE differs significantly in central ten-
dency. A significance level of 0.05 and an adjusted p-value are applied to correct for multiple comparisons (using
the correction after Benjamini & Hochberg (1995)). The results (cf. Figure F1c) demonstrate that SI (best) outper-
forms all "worst" methods and the benchmark-to-beat. However, the null hypothesis for SP and the "best" options
of knn and MLR cannot be rejected. Consequently, rather than identifying a single alternative to the benchmark-
to-beat, we have identified four.
Notably, all regionalization methods lead to poorly performing basins, as evidenced by the range of basins with a
KGE below 0.2, varying from 13 to 37. In Fig. 4c, we examine whether there are basins that all methods cannot

regionalize, thereby indicating a general insufficiency of the regionalization methods for these basins. The histogram indicates that most poorly performing basins belong to a single regionalization method. The high number of basins, which cannot be estimated well by a single regionalization method, illustrates the diverse shortcomings of the methods. A single basin shows poor performance across all methods. This is a basin of the river El Platanito in Mexico. The calibrated $\gamma$ value is about 1.5, and the corresponding KGE value in calibration is 0.466. This basin appears to be highly sensitive to $\gamma$, with an inaccuracy in the estimated $\gamma$ having a significant impact on the accuracy of river discharge. For example, the benchmark-to-beat estimates $\gamma$ to 1.0, which is close to the calibrated value of 1.5. However, the KGE value of the simulated discharge using the benchmark-to-beat is -0.158 due to a high overestimation of the variation and mean of the discharge. This high sensitivity seems outstanding and is likely attributable to the absence of waterbodies and snow, supporting a potentially high impact of $\gamma$ on the model simulation (Kupzig et al., 2023) in conjunction with a relatively small basin size (ca. 6,600 km$^2$).

Model evaluation is at least partially subjective (Ritter & Muñoz-Carpena, 2013), and the choice of evaluation criteria represents a source of uncertainty in model performance evaluation (Onyutha, 2024). Furthermore, the choice should reflect the intended model use (Janssen & Heuberger, 1995). As GHMs are often applied to evaluate monthly simulated discharge (e.g., Herbert and Döll, 2023; Jones et al., 2023; Tilahun et al., 2024), we assess the model performance using monthly data. Moreover, GHMs are generalists rather than expert models; thus, the model evaluation should encompass a range of aspects related to streamflow to obtain an overall metric. Therefore, we applied the monthly KGE, which comprises information about the streamflow's variability, bias, and timing. As we use monthly values, we expect that outliers, i.e., single flood events, are less influential than in daily data sets. Consequently, we expect the disadvantage of the KGE exhibiting sampling uncertainty to be less significant (cf. Clark et al., 2021).

Nevertheless, to reduce the risk that disadvantages of the evaluation criteria influence the model evaluation, we conducted an additional model evaluation using a modified version of the Nash-Sutcliff efficiency (NSE) (Nash & Sutcliff, 1970). This modified NSE uses absolute differences instead of squared terms, leading to a metric that is especially suitable as an overall measure (Krause et al., 2005). The results of the analysis are in Appendix F. The high boxplot similarity between the modified NSE and the KGE confirms that the monthly KGE represents the overall monthly model quality. Moreover, the statistical metrics of the modified NSE indicate that MLR (best), in particular, outperforms the benchmark-to-beat. Applying the one-sided paired Wilcoxon rank sum test on the modified NSE reveals that knn (best), SI (best), and the benchmark-to-beat deliver no statistically significant differences in the central tendency to the well-performing MLR (best). These differences in results illustrate that the choice of evaluation criteria can significantly impact the experimental outcome. Moreover, it underpins the usefulness of evaluating ensemble approaches to account for this inherent uncertainty.

**3.4 Impacts on runoff simulations**

To evaluate the impact of runoff simulations, we apply an ensemble of regionalization methods generating $\gamma$ estimates for the worldwide ungauged regions. Within the ensemble, we use the four methods SI (best), knn (best), MLR (best), and SP that (1) outperform the benchmark-to-beat regarding the logMAE of regionalized and calibrated values and (2) perform similarly to each other and better than the benchmark-to-beat in KGE for monthly discharge. Additionally, we use the benchmark-to-beat as the fifth member of our regionalization method ensemble, as it shows no significantly weaker performance than the well-performing MLR (best) for the modified NSE.

The entire set of 933 gauged basins is used for regionalizing γ, resulting in five distinct worldwide distributions of
γ. The spatially distributed standard deviation of the regionalized values is shown in Fig. 5.
In particular, the southern parts of South America, the northern and southern parts of North America, and Central
Asia reveal differences in γ across the ensemble of regionalization methods (see Fig. 5). In Europe, the highest
differences in regionalized values are observed in Italy, Great Britain, and northern Portugal. In Oceania, the high-
est values in standard deviation of γ are in Tasmania, New Zealand, and the southwest of Australia's coast. In
contrast, a minor variation in γ is apparent in northern Africa, most parts of Australia, and the East of the Dead
Sea. Thus, the uncertainty associated with globally regionalizing γ seems to vary across different regions.

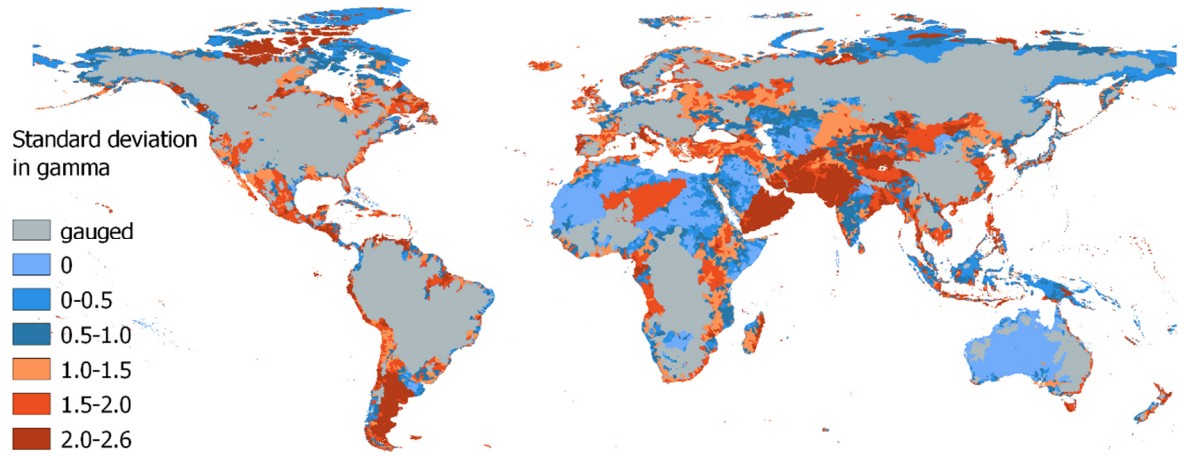

**Figure 5: Standard deviation in regionalized γ values using the best approaches of MLR (best), SI (best), SP, knn (best),**
**and the benchmark-to-beat. Note that dry regions without discharge are set to zero.**
An example of how these uncertainties in regionalized values propagate through the water system is presented in
Fig. 6. This figure displays the coefficient of variation of the mean yearly discharge between 1980 and 2016 based
on the five simulation runs. Moreover, we highlight the effect on rivers in ungauged regions by showing the re-
sulting seasonal pattern, i.e., the simulated long-term mean of monthly river discharge for three exemplary rivers.
These rivers are the Río Bravo in Mexico, the Tiber in Italy, and the Tamar River in Tasmania. Each river is located
in an ungauged region, where the standard deviation in γ is high (see Fig. 5).
Comparing Fig. 5 and Fig. 6 reveals that regions showing variability in γ tend to exhibit variation in mean yearly
discharge. However, the impact of variation in γ on the simulated discharge appears to vary spatially. Some regions
showing a high degree of variation in γ do not exhibit a correspondingly high degree of variation in discharge. For
example, 45 % of all ungauged regions showing a low variation in discharge, i.e., the coefficient of variation is
below 0.5, exhibit a standard deviation of more than one in γ. In contrast, about 89 % of the ungauged regions
showing a higher discharge variation exhibit a standard deviation of more than one in γ. Thus, variation in γ does
not necessarily lead to variation in river discharge, but it increases the likelihood that a region's discharge is af-
fected. The spatially varying impact of γ is likely related to varying sensitivity regarding γ in the ungauged regions,
which depends on numerous aspects, e.g., snow occurrence or waterbodies (see Kupzig et al., 2023).
About 11 % of the ungauged area exhibits variations in yearly river discharge exceeding 50 % of the mean. These
regions are primarily in southern South America and Central Asia. A further 62 % of the ungauged area exhibits
variations in yearly river discharge between 10 % and 50 % of the mean. These regions are mainly located on the
northern coast of Russia and northern Canada, Indonesia, and Tasmania. Other areas, like most ungauged regions
of Africa and Australia, show almost no impact, i.e., the variation in yearly discharge is less than 10 % of the
mean. In northern Africa, one region exhibits higher values in the coefficients of variation. These values are at-
tributable to minimal discharge values, resulting in comparatively high coefficients of variation in this region.
Considering the variation in the seasonality in the selected ungauged river systems (see Fig. 6b-d), the temporal
impact of regionalization varies across the local landscape. For the Tamar River in Tasmania, as illustrated in Fig.
6d, the variation is higher at the start and end of the dry periods in October/November and April/May, respectively.
The spread in monthly mean discharge is about 0.7 m$^3$s$^{-1}$ to 1 m$^3$s$^{-1}$ in these periods. The Tiber in Italy and the Río
Bravo in Mexico exhibit a similar pattern: using the regionalized γ values of SP leads to much higher discharge
rates than other ensemble members, introducing broad uncertainty bands. For the Tiber, this leads to seasonal
estimates varying between 1.2 % (in January) and 11 % (in October) of the mean yearly sum. The Río Bravo shows
variations in its seasonal pattern, with values ranging from 2.2 % (in February) to 6.8 % (in October) of the mean
yearly sum. Thus, all rivers display a temporally varying impact. Whereas the main variation in the discharge of
the Río Bravo and the Tiber is mainly attributed to the SP regionalization run, for the Tamaris River, all regional-
ization runs contribute to the varying long-term monthly mean in discharge.

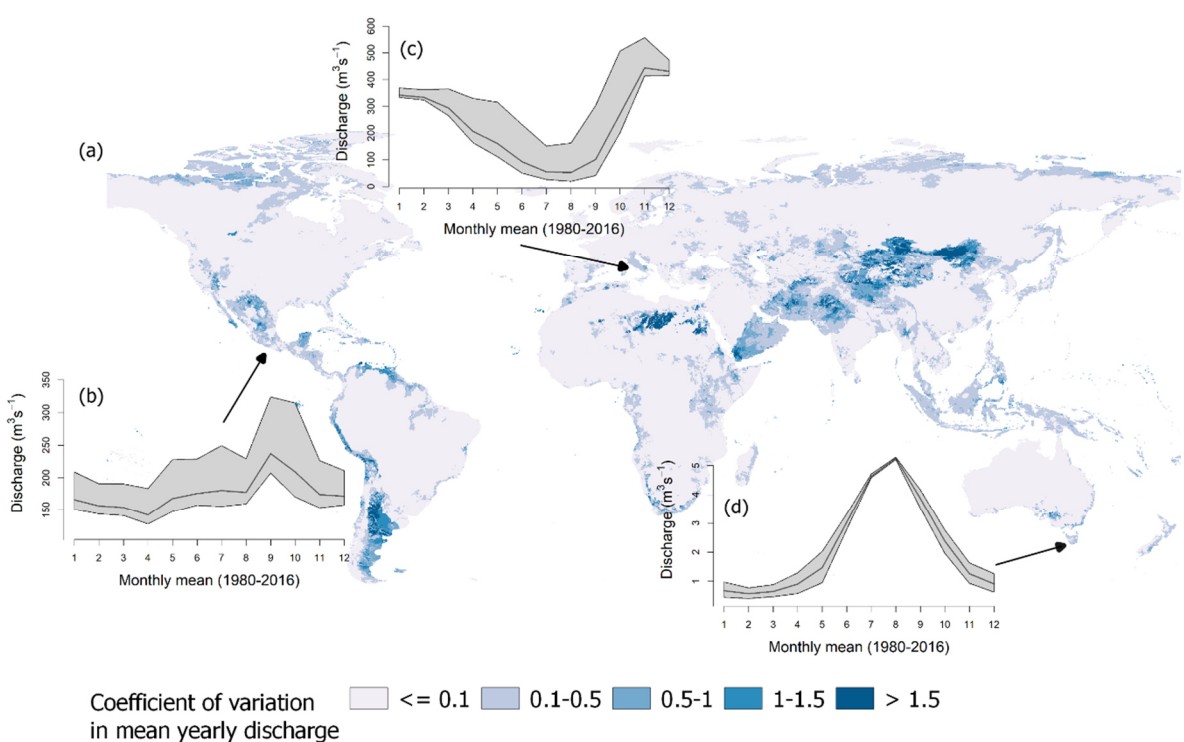


**Figure 6: a) Global map of the coefficient of variation in mean yearly discharge for the applied regionalization methods.**
**Resulting differences in the regionalization ensemble regarding the long-term mean of monthly discharge are depicted**
**for: b) the Río Bravo in Mexico, c) the Tiber in Italy and d) the Tamar River in Tasmania. The grey-shaded area**
**indicates the range of the long-term mean of monthly discharge and the black line indicates the mean off all simulation**
**runs.**
To gain a deeper understanding of the local impact of regionalization on runoff simulations, we analyze the annual
percentiles from 1980 to 2016 for Río Deseado in Argentina, Río Bravo, and Tamar River, displaying the mean
percentile of all years (see Fig. 7a-c). As the Tiber and Río Bravo display high similarities in the resulting patterns
of percentiles, we demonstrate the impact by showing the percentiles from the Río Bravo. Additionally, we com-
pare the relative differences in the mean for each percentile using eight ungauged river systems (see Fig. 7d), as
previously done by Gudmundsson et al. (2012) for nine GHMs. To calculate the relative difference, we subtract
the mean annual percentile of a method from the corresponding mean annual percentile of the reference and divide
the resulting difference by the mean annual percentile of the reference. Instead of using observed flow as a refer-
ence, we use the annual percentiles of our benchmark-to-beat. As river discharge is already spatially aggregated
information, it is unnecessary to spatially aggregate grid cells to create results comparable to those of Gudmunds-
son et al. (2012), who used cell runoff. The evaluated river systems are Río Chubut, Río Deseado, Río Negro, Río
Bravo, Tamar River, Tiber, Pescara, and Ebro.

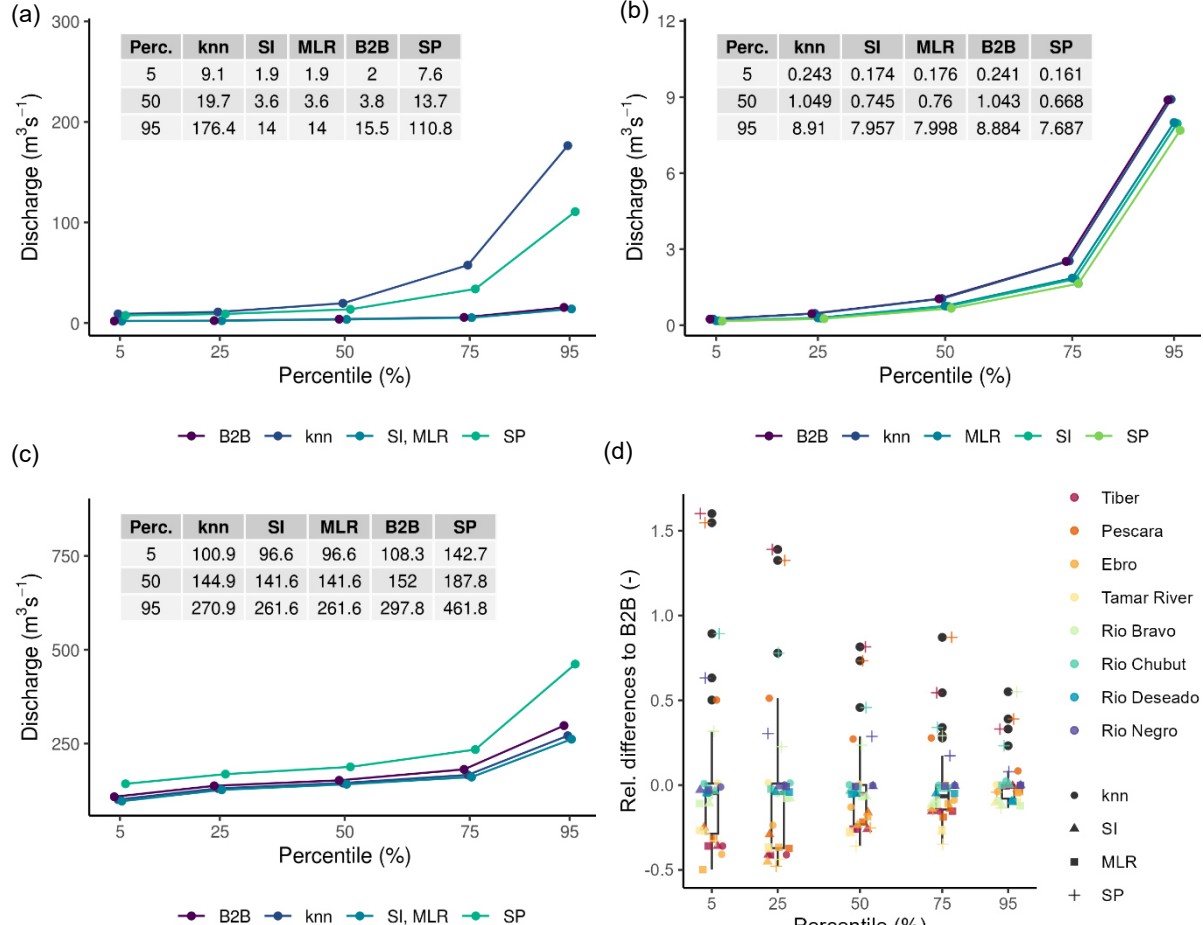

**Figure 7: Mean annual percentiles between 1980 and 2016 of simulated discharge using an ensemble of regionalization**
**methods. The rivers are a) Río Deseado, b) Tamar River, and c) Río Bravo. In d), the relative differences in mean annual**
**percentiles to the benchmark-to-beat of eight ungauged river systems are presented. Negative values indicate smaller**
**mean annual percentiles than the benchmark-to-beat. Note that all data points from Río Deseado for knn and SP are**
**excluded as the values are above 2.0.**
In Fig. 7a, Río Deseado is highly affected by uncertainties in simulated discharge due to the different regionaliza-
tion methods; all segments of the percentiles show high variations where the absolute spread is increasing with
increasing percentiles. For SP and knn (best), the discharge is highest, e.g., estimating a median discharge of 13.7
$m^3s^{-1}$ and 19.7 $m^3s^{-1}$, respectively. For the other methods, the simulated discharge is low, e.g., SI and MLR result
in an equal median discharge of 3.6 $m^3s^{-1}$. The Tamar River in Fig. 7b also shows increasing absolute differences
between the methods for higher percentiles, with the benchmark-to-beat approach leading to the highest discharge.
For the Río Bravo, the absolute differences between the highest result of SP and the other methods remain almost
constant until the 75[th] percentile. For the 95[th] percentile, the absolute differences increase rapidly from about 40
$m^3 s^{-1}$ (75[th] percentile) to nearly 200 $m^3 s^{-1}$ (95[th] percentile). The exemplary results of Río Deseado and Río Bravo
indicate a potentially high degree of uncertainty regarding the high percentiles in discharge simulation. These
uncertainties put the results of global flood frequency analysis (e.g., Ward et al., 2013) in ungauged regions at risk
as the time series of annual maxima might be even more uncertain. Thus, the results of flood frequency analysis
should be carefully interpreted in ungauged regions as the impact of parameter regionalization may be significant.
Upon examination of the relative differences to the benchmark-to-beat for eight ungauged river systems, it be-
comes evident that the impact of regionalization methods varies between ungauged river systems (e.g., Río Negro
exhibits almost no variation, but Ebro does). Moreover, it becomes apparent that some regionalization methods
contribute more to the variation in estimated discharge than others. The methods contributing most are knn (best)
and SP. For knn (best), 10 of the 40 relative differences are higher than |0.3|. For SP, even 29 out of the 40 relative
differences are higher than |0.3|. The results of SI (best) and MLR (best) are very similar, indicating high similarity
in performance. This is consistent with the KGE evaluation (see Chapter 3.3), in which they performed similarly.
The observation in Fig. 7d that higher relative differences of discharge simulations occur in drier percentiles is
also reported in Gudmundsson et al. (2012). Moreover, the relative differences between the five regionalization
runs seem comparable to the inter-model differences depicted in Gudmundsson et al. (2012), indicating the high
impact of regionalization methods on the evaluated ungauged river systems.
Finally, Table 3 presents the estimated yearly mean runoff to the ocean for all five ensemble members. All esti-
mates of global "runoff to ocean" range from 45,622 (SI (best)) to 47,069 (SP). Thus, the differences are on the
scale of smaller inter-model differences (see Table 2 in Widen-Nilsson et al.,2007). The impact of regionalization
becomes even more evident using an unsuitable regionalization method for WaterGAP3. For instance, the tuned
kmeans ("subset") approach results in 42,862 $km^3 yr^{-1}$ "runoff to ocean", increasing the spread between the meth-
ods to 4,208 $km^3 yr^{-1}$ being in the scale of inter-model differences. This high impact of regionalization on global
"runoff to ocean" is surprising, given that only 27 % of the world is ungauged, using the GRDC database. From
this 27 %, most regions are in Australia and Africa, where minimal runoff is produced. In studies employing
disparate models, e.g., for inter-model comparison, all regions are simulated in disparate ways.
**Table 3: Mean outflow to the ocean and endorheic basins in km³ yr⁻¹ between 1980-2016. The highest continental devi-**
**ation to the benchmark-to-beat is indicated in bold.**

| Runoff to ocean[1] | B2B | SI (best) | knn (best) | MLR (best) | SP |
|---|---|---|---|---|---|
| Oceania | 1,127 | -1.80 % | -2.20 % | -3.40 % | **-6.60 %** |
| Europe | 3,098 | -2.30 % | -0.10 % | **-2.60 %** | 0.20% |
| Asia | 16,676 | 3.50 % | 0.30 % | 1.60 % | **5.50 %** |
| Africa | 5,203 | -1.00 % | 0.70 % | -0.30 % | **-3.60 %** |
| North America | 7,517 | 0.30 % | 1.00 % | -1.70 % | **2.20 %** |
| South America | 12,032 | 1.30 % | 1.40 % | -0.20 % | **4.90 %** |
| global | 45,653 | 46,273 | 45,953 | 45,622 | 47,069 |

[1]including endorheic basin


The most significant deviations in the continental sums of "runoff to ocean" in Table 3 are due to SP. Only for
Europe is the highest deviation related to MLR (best), not SP. Interestingly, the estimated sums of SP occasionally
define the lowest and occasionally the highest extremes for the continents, lacking a systematic pattern. The out-
standing role of SP is consistent with previous evaluations in this Chapter, where SP frequently contributes most
to the variation in discharge. This suggests that SP may not be suitable for the global scale. Nevertheless, the
pseudo-ungauged basins in the split-sample tests may also exhibit considerable distances from the observed basins.
Given that SP achieved satisfactory results in both evaluations, using either the logMAE or the KGE, the evaluation
indicates the method's suitability on a global scale. Thus, in the future, the split-sample test must be extended to
gain deeper insights into the method's robustness and make a definitive statement about the method's suitability
on a global scale. For example, the so-called "HDes" approach, recommended by Lebecherel et al. (2016), could
be applied for this purpose. In this approach, the closest basin to the corresponding (pseudo-) ungauged basin is
excluded from the regionalization process, thereby enabling an assessment of the method's robustness.
**3.5 Challenges & Future Directions**
Regionalization is an inevitable step when parameterizing GHMs. However, only a few studies exist that conduct
regionalization experiments with GHMs, often focusing on a single or two distinct regionalization strategies (e.g.,
Beck et al., 2016; Beck et al., 2020; Yoshida et al., 2022). A significant challenge in developing and testing dif-
ferent regionalization methods for GHMs is the time-consuming runtime of these models. This extensive runtime
impedes comprehensive testing of different regionalization methods, as evaluating the regionalization methods,
e.g., by using streamflow, demands a considerable number of simulation runs. This study addressed this challenge
using the differences between calibrated and regionalized parameter values as an approximator for the suitability
of the regionalization methods. Thereby, we considered the varying sensitivity of the parameter within the param-
eter space using the logMAE as the evaluation criterion. Using the differences between calibrated and estimated
values is the most straightforward approach, given that WaterGAP3 uses a single calibration parameter, leading to
a clear global optimum. However, this approach might not apply to GHMs using multiple calibration parameters
due to equifinality. For example, Ayzel et al. (2017) found varying estimated parameter values when regionalizing
11 parameters of the SWAP model using different regionalization methods. They concluded that the difference
between regionalized and calibrated values cannot be regarded as a performance measure due to parameter com-
pensation. Thus, further research is required to tackle the challenge of GHMs' time-consuming runtimes to enable
comprehensive testing of regionalization methods, especially for GHMs using multiple calibration parameters.
Another challenge in regionalizing hydrological models is the optimal selection of predictors for the regionaliza-
tion methods. Various approaches exist regarding the predictor selection for the regionalization methods (Razavi
& Coulibaly, 2013), resulting in a lack of consensus. This study used a predictor selection based on correlation
coefficients and an entropy assessment. The results indicate that the approach is particularly well-suited to the
Similarity Indices. However, further research on predictor selection is needed to find the optimal descriptor set
per method, as regionalization methods use predictors with varying efficiency. For example, future studies might
integrate feature importance bars, e.g., by using permutation, to identify the most critical descriptors per method.
Moreover, future research should explicitly account for the issue of multicollinearity. Multicollinearity can affect
MLR (and potentially other techniques), resulting in ungeneralizable predictions. This phenomenon is more
likely to occur when the number of predictor variables is large relative to the number of observation units and
when the predictor variables are highly collinear (Kiers & Smilde, 2007). To account for the high importance of
the generalizability of regionalization methods for GHMs, we used a high proportion of the basins for testing,
i.e., 50 %. Moreover, we used a large sample size (50 % of 933 basins) relative to the number of predictors
(maximum 12), lowering the risk of multicollinearity interfering with the results. However, future studies might
use methods such as Principal Component Analysis (PCA) or Partial Least Square (PLS), explicitly accounting
for the issue of multicollinearity (e.g., Kroll & Song, 2013). An alternative approach to using PCA or PLS is ex-
plicitly testing for multicollinearity in predictor sets using the variance inflation factor and avoiding using pre-
dictors with values exceeding a pre-defined threshold (e.g., Kroll et al., 2004).

## 4. Conclusion

Valid simulation results from GHMs, such as WaterGAP3, are crucial for detecting hotspots or studying patterns
in climate change impacts. However, the lack of worldwide monitoring data makes adapting GHMs' parameters
for valid global simulations challenging. Therefore, regionalization is necessary to estimate parameters in un-
gauged basins. This study applies regionalization methods for the first time to WaterGAP3, aiming to provide
insights into selecting suitable regionalization methods and evaluating their impact on the runoff simulations. Tra-
ditional and machine learning-based methods are tested to assess the application of several regionalization tech-
niques on a global scale. The concept of benchmark-to-beat and an ensemble of split-sampling tests are employed
for a comprehensive evaluation. Moreover, the impact on runoff simulation is assessed using a wide range of
temporal and spatial scales, i.e., from the daily to the yearly and from the local to the global scale.
In this study, four regionalization methods outperform the benchmark-to-beat in monthly KGE and are thus con-
sidered appropriate for WaterGAP3. These methods span the complete range of methodologies, i.e., regression-
based methods and methods using the concept of physical similarity and spatial proximity. Moreover, the methods
vary in the descriptors used to achieve the highest accuracy. This highlights that different methods use descriptor
sets with varying efficiency. All methods perform best when using climatic and physiographic descriptors, indi-
cating that combining climatic and physiographic descriptors is optimal for regionalizing worldwide basins.
Mainly for two selected regionalization methods (tuned MLR and knn), the suggested descriptor selection based
on correlation coefficients and entropy assessment is not optimal. Further research might integrate variable im-
portance scores or PCA to enhance the predictor selection. Although random forest is known to be especially
robust among other machine learning-based techniques, it shows symptoms of over-parameterization, indicating
that the algorithm is too flexible and adjusts to noise in the data, missing the underlying systematic pattern.
Our results demonstrate that variation in the regionalized parameter value does not necessarily lead to variation in
river discharge. However, it increases the likelihood that a region's runoff is affected. This spatially varying impact
of $\gamma$ is likely related to the varying sensitivity in ungauged regions regarding $\gamma$. Southern South America is a region
identified to be especially sensitive to variation in $\gamma$. Furthermore, local effects on runoff simulations indicate a
temporally varying impact. For example, some impacted rivers indicate a high degree of uncertainty regarding the
high percentiles in discharge simulation. These uncertainties potentially lead to a significant impact on flood fre-
quency analysis on a global scale, where the lack of gauging stations in certain regions calls for regionalization.
The global impact of regionalization methods that perform well for WaterGAP3 appears to be in the order of minor
inter-model differences. This impact rigorously increases when using a poorly performing method for WaterGAP3,
underscoring the importance of carefully selecting regionalization methods.
The spatial proximity approach contributes most to the variation in estimated runoff. The outstanding role of this
approach suggests that it may not be suitable for the global scale. However, as the pseudo-ungauged basins in the
split-sample tests may also have considerable large distances to the observed basins, and the method achieves
satisfactory results in all executed evaluations, it is not possible to make a definite statement about the method's
suitability for the global scale. Further research is required to gain deeper insights into the methods' robustness,
e.g., by extending the analysis by applying the recommended "HDes" approach (Lebecherel et al., 2016).
*Code and data availability.* The data and the supporting R-Code to reproduce this study's findings are available at
https://doi.org/10.5281/zenodo.12808527.
*Authors contribution.* JK developed, designed, and drafted the study. NK helped to design the experiment. MF
provided feedback throughout the entire process and supported the writing.
*Competing interests.* The authors declare that they have no conflict of interest.
**Appendix A: Global Map of derived global soil moisture storage**

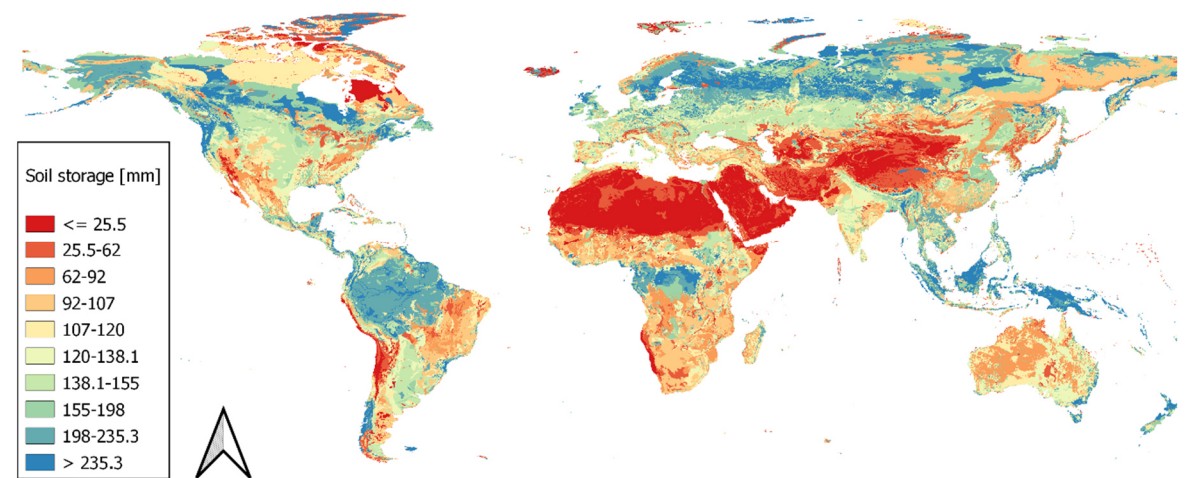


**Figure A1: Global map of the size of soil storage based on Batjes (2012) and land use information (derived from Friedl**
**& Sulla-Menashe, 2019)**

**Appendix B: Further analysis regarding the clustering of parameter values at the extremes**
The clustered calibrated parameter values at the extremes of the valid parameter space (see Fig. 1b) are a known
problem within the calibration. As the parameter space, i.e., the parameter bounds, is crucial for calibration and,
in consequence, for regionalization, we address this issue by a brief sensitivity analysis to demonstrate that the
clustering of the calibrated parameter values is more an issue of missing processes (or using additional parameter
values) than an issue of inappropriate parameter space. As the lower limit of the calibrated parameter (0.1) is
sufficiently small in comparison to other studies using a similar HBV-based approach for runoff generation pro-
cesses (e.g., see the beta in Table A2 in Jansen et al., 2022), we focus on the sensitivity analysis on the upper limit
of γ (5.0).
In the sensitivity analysis regarding the upper limit of γ, we applied the model formula (see equation B1) containing
the model's parameter γ and modified it within the bounds of 0.1 and 10. Additionally, we modified the soil satu-
ration varying from 1 % to 95 %.

$$outflow = precipitation_{effective} \cdot soil\ saturation^{gamma} \qquad\qquad\text{(B1)}$$

The calculated outflow and its relationship to the soil saturation and γ are depicted in Fig. B1 and B2. The incoming
effective precipitation is defined as constant. As it is a factor in equation B1,, the results regarding incoming
effective precipitation are linearly scalable.

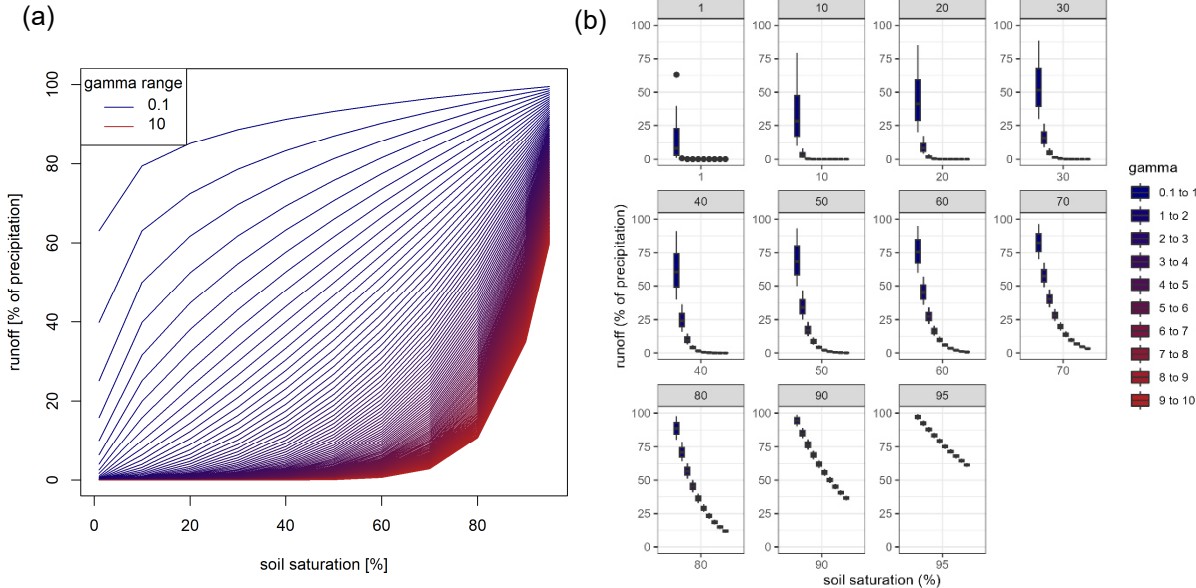

**Figure B1: a) Runoff generation in the soil layer (neglecting overflow and evapotranspiration) using different values**
**for the calibration parameter and increasing the soil-moisture, b) runoff generation for varying soil moisture grouped**
**in bins of size one.**
In the depicted Fig. B1, the runoff generation process differences between differing γ values become more linear
when soil saturation increases. Thus, the non-linear model parameter becomes less critical for high soil moisture.
Generally, the runoff generation process differences for higher γ values are more pronounced for higher soil mois-
ture. For lower soil moisture, the smaller values have higher effects on the generated runoff. For example, for 70 %
soil moisture, the differences for γ values ranging from 5 to 10 are between 3 % and 16 %. For the same soil
moisture, the range in runoff generation varies from 16 % to 70 % for γ values between 1 and 5.
High γ values usually occur in dry regions (see Fig. 4b in Müller Schmied et al., 2021). In dry regions, high soil
moisture values are not expected to occur frequently (e.g., see Khosa et al., 2020; Oloruntoba et al., 2024 for
estimated and measured soil moisture in Africa and Draper et al., 2008 for estimated and measured soil moisture
in Australia). It is, therefore, unlikely that higher γ values will significantly enhance the calibration result or de-
crease the issue of clustered calibrated parameter values at the higher end of the parameter space. More likely, the
clustering of calibrated parameter values will be resolved in dry regions by incorporating additional (missing)
model processes, such as evaporation from rivers or inaccurate representation of groundwater processes (Eisner,
2016, p. 49). Thus, the parameter bounds of γ (e.g., also used in Eisner 2016, p. 16; Müller Schmied et al., 2021;
Müller Schmied et al., 2023) are not changed in this study.

**Appendix C: Basin descriptors**

Overview of basins descriptors used in this study. All basin descriptors are derived from the original model input and aggregated with a simple mean method to basin values to produce the same spatial resolution as the calibrated model parameter.

- *Soil Storage*: The size of the soil storage, i.e., the maximal water content in the soil reachable for plants in mm. The information is the product of rooting depth (defined in a look-up table) and the total available water content derived from Batjes (2012).

- *Open Water Bodies*: The fraction of the area covered with open water bodies in the basin is given as a percentage. The model input is based on the GLWD database (Lehner & Döll, 2004).

- *Wetlands*: The fraction of area covered with wetlands in a basin is given in percentage. The model input is based on the GLWD database (Lehner & Döll, 2004).

- *Size*: Size of a basin in km$^2$.

- *Slope*: The mean slope class is calculated as described in Döll & Fiedler (2008) and based on GTOPO30 (USGS EROS data centre).

- *Altitude*: The mean altitude of a basin is given in meters above sea level and based on GTOPO30 (USGS EROS data centre).

- *Forest*: The mean fraction of the area covered with forest is given in percentage and derived from MODIS data (Friedl & Sulla-Menashe, 2019), where 2001 is used as a reference. All grid cells having a dominant International Geosphere-Biosphere Programme (IGBP) classification between one and five are defined as "forest".

- *Sealed Area*: The mean fraction of sealed area is given in percentage and derived from MODIS data (Friedl & Sulla-Menashe, 2019), where 2001 is used as a reference. All grid cells having an IGBP classification equal to 13 are defined as they would contain 60% of the sealed area. Note: The different treatment of forest and sealed area is based on the required model input; whereas the land cover is a classified value, the sealed area is a floating-point value.

- *Permafrost & Glacier*: The mean coverage of permafrost and glacier in a basin is given in percentage. It is based on the World Glacier Inventory and the Circum-Arctic Map of Permafrost and Ground-Ice Conditions.

- *Mean Temperature*: The mean air temperature is based on the meteorological forcing used to drive the model (Lange, 2019) covering the period 1979 to 2016 and given in degrees Celsius.

- *Yearly Precipitation*: The yearly precipitation sum is based on the meteorological forcing used to drive the model (Lange, 2019) covering the period 1979 to 2016 and given in mm.

- *Yearly Shortwave Downward Radiation*: The yearly shortwave downward radiation is based on the meteorological forcing used to drive the model (Lange, 2019) covering the period 1979 to 2016 and given in Wm$^{-2}$.

The correlation between the defined basin descriptors is shown in Fig. A1. The variation within each basin descriptor for basins used for regionalization is shown in Fig. A2.

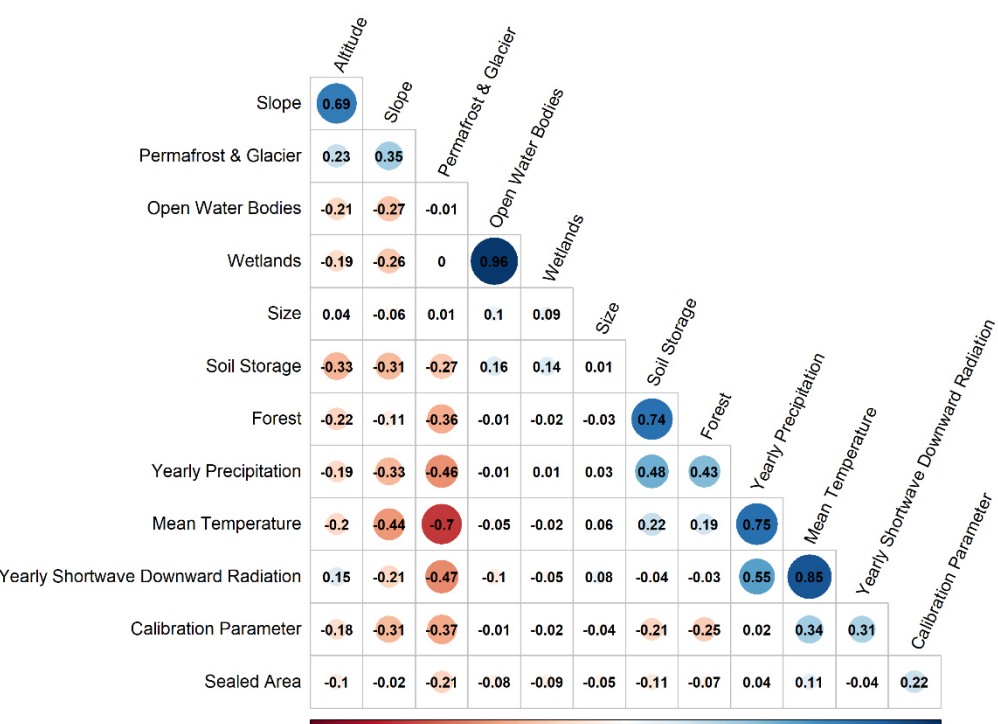


**Figure C1: Correlation (using Pearson's correlation) between basins descriptors.**


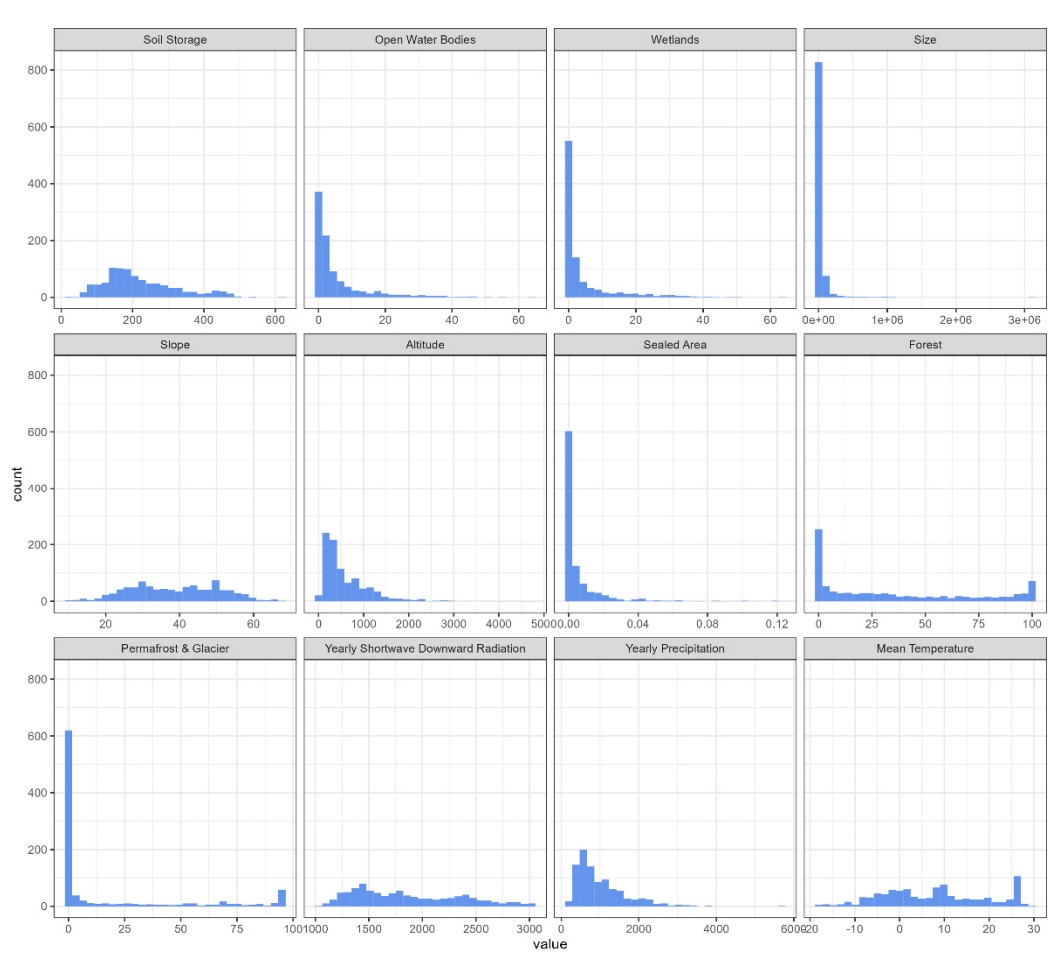


**Figure C2: Distribution of basins descriptors within all basins used for regionalization (n=933)**

**Appendix D: Results of the ensemble of the split-sample tests**

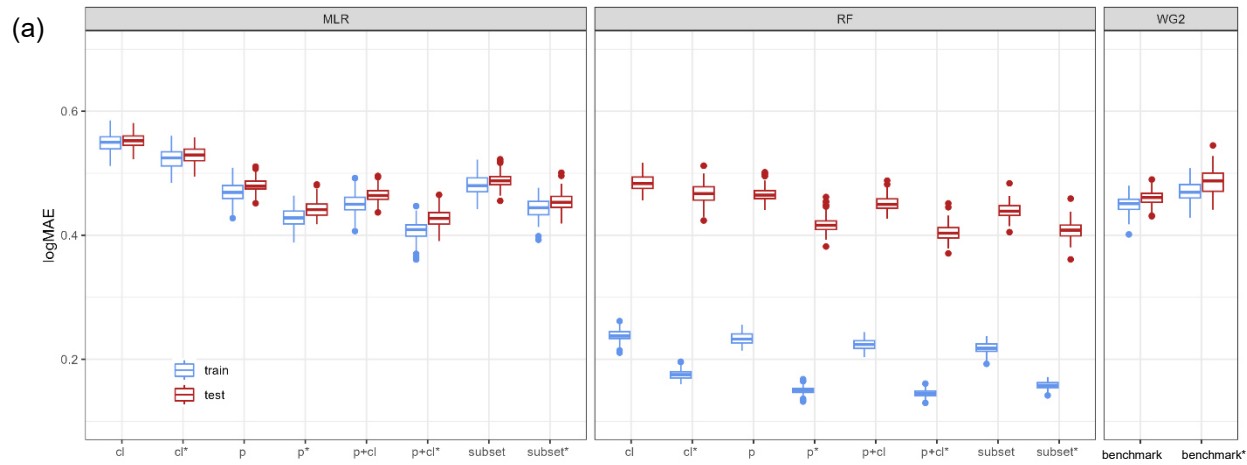


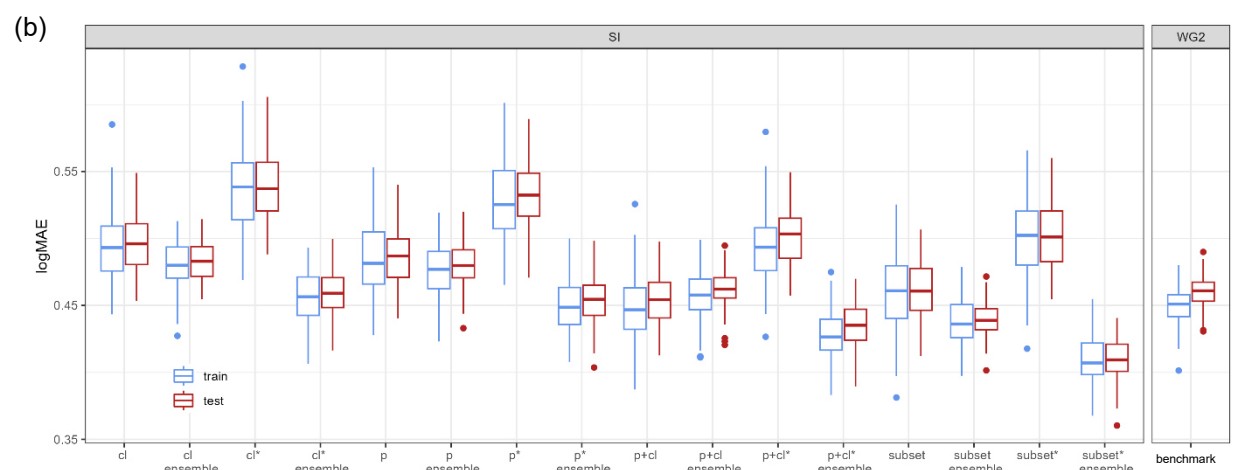


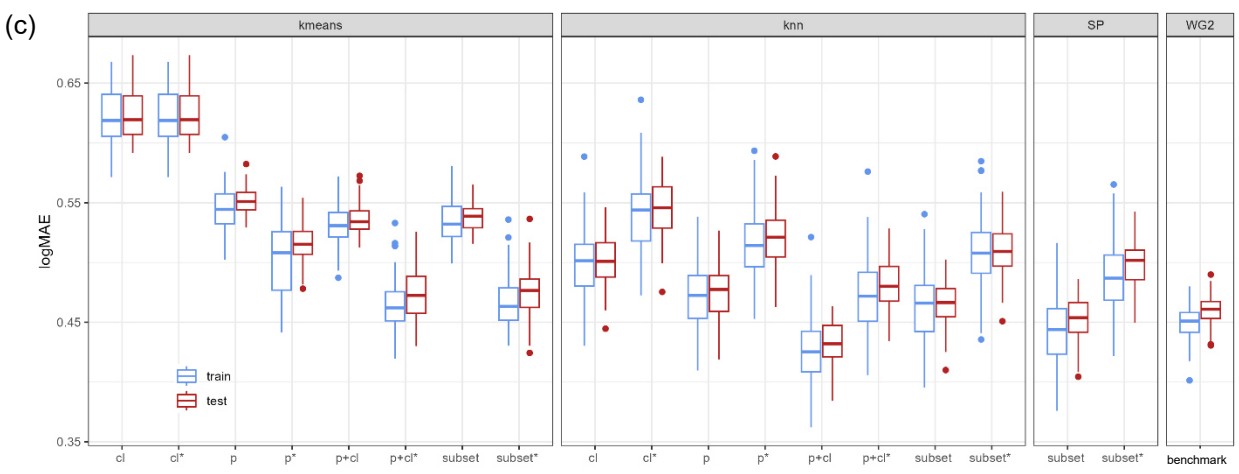


**Figure D1: logMAE values for all 100 split-sampling tests using all variants of a) MLR, RF, and benchmark-to-beat,**
**b) SI, and c) kmeans, knn, and SP. Note that the asterisk * indicates the tuned version of the method.**

**Table D1: Performance loss in median logMAE of the ensemble of split-sample tests from training to testing expressed**
**in % of logMAE in training.**

| test (% train) | MLR | RF | SI | | kmeans | knn | SP | B2B |
| --- | --- | --- | --- | --- | --- | --- | --- | --- |
| | | | no ens. | ensemble | | | | |
| cl | 100.4 | 202.9 | 100.6 | 100.6 | 100 | 100 | | |
| p | 102.1 | 199.6 | 101.2 | 100.6 | 101.3 | 101.1 | 102.3 | 102.2 |
| p+cl | 103.1 | 207.1 | 101.6 | 100.9 | 100.6 | 95.6 | | |
| subset | 101.7 | 223.9 | 100 | 100.7 | 101.3 | 100.2 | | |

| test* (% train*) | MLR | RF | SI | | kmeans | knn | SP | B2B |
| --- | --- | --- | --- | --- | --- | --- | --- | --- |
| | | | no ens. | ensemble | | | | |
| cl | 100.8 | 266.9 | 99.8 | 100.7 | 100 | 100.4 | | |
| p | 103 | 277.3 | 101.3 | 101.3 | 101.4 | 101.4 | 103.1 | 104.1 |
| p+cl | 104.4 | 277.9 | 102 | 102.1 | 102.2 | 101.7 | | |
| subset | 102 | 258.2 | 99.8 | 100.5 | 103 | 100.2 | | |



**Appendix E: Feature importance bars for MLR (best) and knn(best) using the descriptor set "p+cl"**


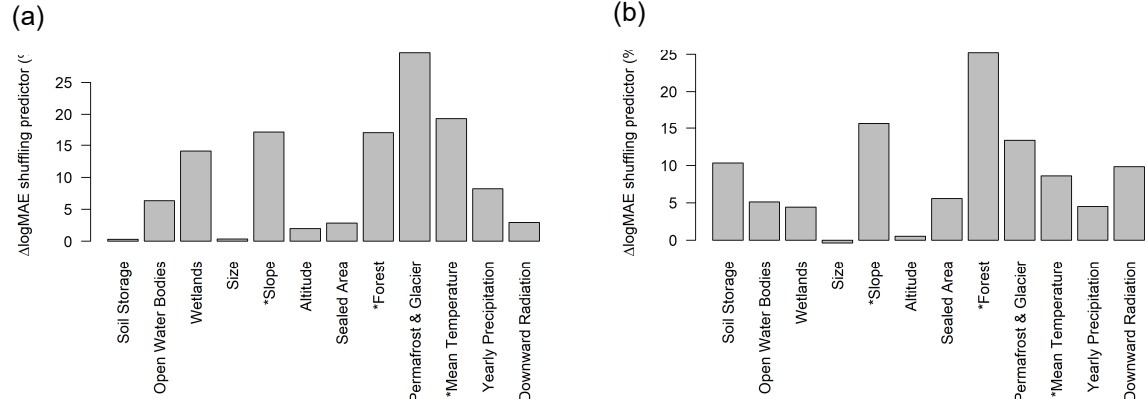


**Figure E1: Decrease in logMAE for testing using one representative split-sample when randomly shuffling each pre-dictor for a) MLR (best) and b) knn (best). Note that the asterisk indicates the basin descriptors used in the (weakly) correlated subset.**


 **Appendix F: Model performance for pseudo-ungauged basins using a modified version of the NSE**

Krause et al. (2005) suggested a modified version of the NSE that is especially suitable as an overall metric, leading
to results between NSE versions focusing on low and high flows. The applied equation for the modified version is
given below (see Eq. F1).
$modified\ NSE = 1 - \frac{\sum|y_k - x_k|}{\sum|y_k - \mu_y|}$ *(F1)*
where $x_k$ is the simulated monthly discharge for the timestep $k$ and $y_k$ is the observed discharge for the timestep
$k$, and $\mu_y$ is the mean of the discharge for the evaluated period.
The evaluation of the modified NSE for all pseudo-ungauged basins of a representative split-sample are summa-
rized in Figure F1. Note that the figure includes also the results of the applied one-sided paired Wilcoxon rank
sum test for the KGE values, mentioned in Section 3.3.

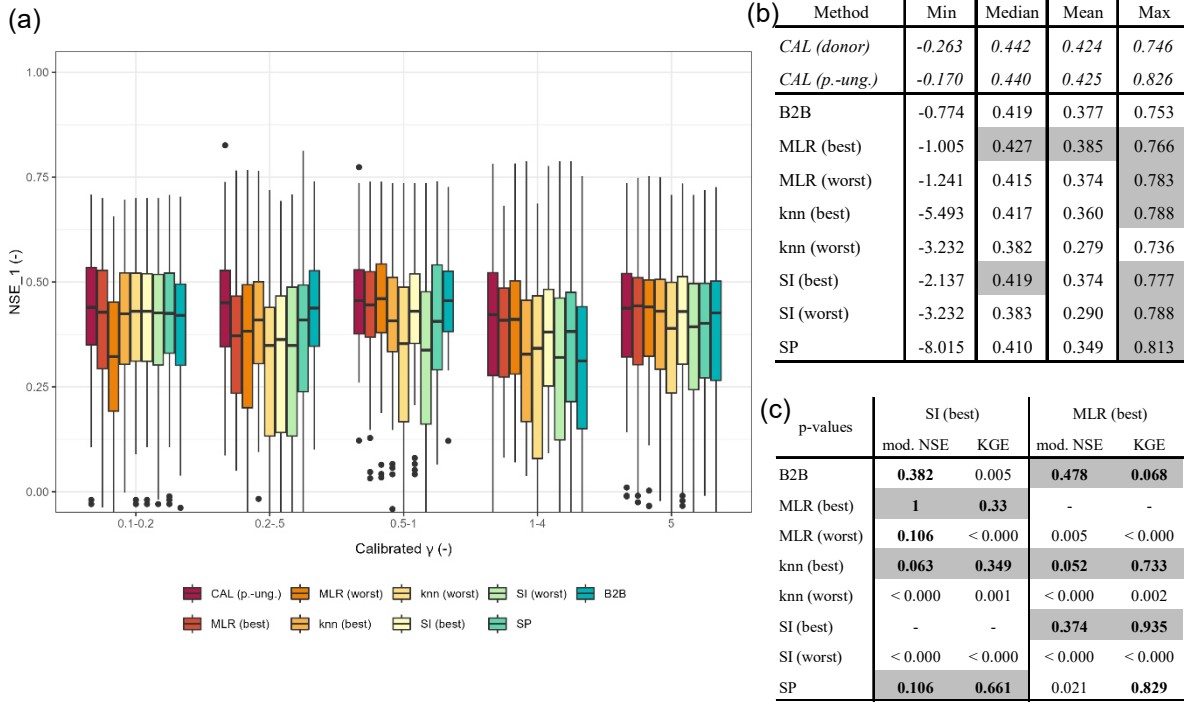

**Figure F1: a) modified NSE values of pseudo-ungauged basins from split-sample test grouped by the range**
**of calibrated γ values, b) selected metrics of modified NSE values from the pseudo-ungauged basins (bet-**
**ter or equal performance to the benchmark-to-beat is highlighted in grey), and c) p-values of the one-sided**
**paired Wilcoxon rank sum test, testing the best performing methods MLR (best) and SI (best) against all**
**other regionalization methods. (Note that p-values greater than 0.05 are highlighted in bold, indicating**
**that the null hypothesis cannot be rejected, thus the difference in central tendency is not statistically sig-**
**nificant; cases where the results of modified NSE and KGE indicate the same are shaded grey.)**

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
