# Peer review of "Regionalization in global hydrological models and its impact on"

_Geoscientific Model Development, 2024_

## Author Comment (AC3)

**Response to Reviewer 1**

We thank the referee for the helpful review and the constructive comments. His/her comments are given below in black, and our responses are in blue. Proposed changes to the manuscript are in red. Additional references are provided at the end of this document.

The article presents a comparison of regionalization methods to estimate the calibration parameter of the WaterGAP3 GHM model in ungauged catchments worldwide. The article presents relevant work, and it is well written. Although, in my opinion, the fact that the methodology relies completely for data and evaluation in the model rest validity to analysis and findings of the results. Nevertheless, if these constraints and their implications are better analyzed and addressed, the paper is publishable.

Thanks for addressing the omission of constraints and implications that may affect the reliability of the findings. To address this issue, we modified the paper by (1) explaining choices better (e.g., why we use basin descriptors derived from model input data), (2) addressing underlying assumptions (e.g., using the difference between calibrated and regionalized values to assess the performance of regionalization by adding a KGE evaluation for one split-sample), and (3) restructuring the result section and creating more convincing figures that address more precisely the difference between the regionalization methods and the impact on run-off simulations.

**General comments:**

In several parts of the manuscript, the authors claim they are using new regionalization methods, and that is misleading, they are applying regionalization methods (that are not new) for the first time in the context of the WaterGap model. Please clarify throughout the text.

Thanks for the precise comment. We have clarified throughout the text that the context of the applied methods, i.e., the application for WaterGAP3 on a global scale, is new, but not the methods themselves.

The implementation of regionalization methods is not very well explained, many assumptions and choices are made without proper justification (see specific comments). These may have important impacts in the performance of each regionalization method, unfairly favoring one over another.

Thanks for the valuable, specific comments regarding this issue. We have addressed the comments in detail in the corresponding 'specific comment' section. We modified the text to clarify the assumptions and choices. Moreover, we have modified the 'highly-flexible' kmeans method as (1) the number of clusters was not according to any benchmark and (2) directly using knn to find the most similar donor basins enables an improved assessment of the benefit of using clusters.

I have some reservations with the fact that all regionalization is made within the WaterGAP model, e.g.,

- the use of model input data as basin descriptors, for example "soil storage" coming from a look-up table, does not seem to be a proper descriptor of a basin;

   Thanks for raising this issue. We understand that the input for regionalization is critical and appreciate that it is highlighted. As we need (rasterized) globally available data for regionalization, we have decided to use model input as it is already available on the needed spatial resolution. We did so, as the used model input is mainly based on global data products that are mostly only adapted to fit the spatial resolution (e.g., waterbody information, digital elevation map). The "soil storage" differs from that as it is a combination of three pieces of information: soil information derived from Batjes (2012), land use information derived from MODIS data (Friedl & Sulla-Menashe, 2019), and estimated rooting depth coming from a look-up table (used to match the defined land use classes from MODIS data). We could have used all three pieces of information separately, and we did so partly as we integrated the mean fraction of forest to account for different land use information. However, as we noticed that the soil storage size significantly influenced the simulation during manual simulation runs, we anticipated similar effects for regionalization. Therefore, we decided to directly implement the estimated soil storage size as a basin descriptor. However, to enhance the study's reproducibility and reinforce the findings' reliability, we added the calculated soil storage size as a map in the Appendix A and the provided the data as .tiff in the Zenodo repository.

- the selection of these descriptors based on the correlation with the calibrated parameter;

Thanks for mentioning this aspect and providing food for thought. As we estimate only one parameter, the most straightforward approach is using the correlation between this parameter and the potential descriptors (also done by others, even for more than one calibrated parameter, e.g., Merz & Blösch, 2004; Wagener & Wheater, 2006). However, it should be acknowledged that alternative approaches exist, e.g., using flow characteristics (e.g., Pagliero et al., 2019), which aim to relate basin descriptors to hydrological processes and, thus, model behavior.

Given the low correlation coefficients between the descriptors and calibrated parameters, we proceeded to modify the analyzed subset to reduce the impact of the selection on the study. Therefore, we run the regionalization methods with all descriptors additionally to evaluate the descriptor selection. Furthermore, to ascertain the advantage of integrating climatic descriptors, we run the regionalization methods using all physiographic or all climatic descriptors.

Additionally, the discussion of the low correlation coefficients is extended by presenting the results in the context of other studies.

- the evaluation of each method by comparing regionalized gamma versus calibrated gamma (if I understood correctly);

As WaterGAP3 is very demanding in terms of computational time, it is advisable to take advantage of the calibration result corresponding to the clear global optimum of the single calibration parameter. So, being closer to the calibrated value means becoming closer to the global optimum and thus better in regionalization. However, as the sensitivity of the calibration parameter changes between basins and depends on the parameter value itself, we admit there is the need to validate the assumption that the differences between calibrated and regionalized values result in differences in the simulation. Therefore, we (1) formulate the underlying assumption clearly in the text and (2) include an evaluation of KGE values of a representative split-sample to validate it. Moreover, when assessing the differences between regionalized and calibrated values, we changed the MAE to logMAE to account for the increasing sensitivity for smaller parameter values. To show this change in sensitivity, we added an Appendix section (see Appendix B) presenting a brief sensitivity analysis regarding the calibration parameter.

- and also, to use as benchmark the resulting parameter distribution of a previous version of the model at a different resolution and regionalization method.

We are sorry that it was not clear that the benchmark is not the parameter distribution of WaterGAP2 itself but only the regression equation. This regression equation was fitted to our data and defined as benchmark-to-beat. We clarified this in the text.

**Specific comments:**

Line 131, explain the standard calibration approach and sufficient performance indicators referred here.

Thanks for highlighting that mentioning the 'standard calibration' was not sufficiently indicating that it refers to the calibration procedure described in ll. 97-107. To clarify this, we added the term 'above-described' (ll. 133). Moreover, we added information about the implemented search algorithm to the calibration description (ll. 99-107).

Further, to create more valid results, we implemented one additional performance criterion (monthly KGE $\geq 0.4$) to select gauged basins used in this study, reducing the number of examined basins from formerly 1,236 to 933. (s. ll. 142-151)

Line 148-149, explain what do you mean with the need to define sensitive parameters for a better calibration of WaterGAP3 if the model only has one calibration parameter.

Thanks for highlighting the difficulty of understanding this statement. To enhance the clarity, we have revised the paragraph to be more comprehensible (ll. 149-160). In particular, we added the following explanation to answer the raised question: "Future studies are needed to achieve the latter, as WaterGAP3 contains many hardcoded parameters or parameters defined by look-up tables that need to be analyzed to identify and adjust sensitive parameters more accurately during calibration" (ll. 157-159).

Line 162-163: I would say you are minimizing redundant information, but not avoiding it, some of the descriptors are correlated.

Thanks for the suggestion; we applied it to the text and changed 'avoid' to 'minimise' (ll. 175).

Line 173 to 177: To use the calibrated parameter to select descriptors comes with many drawbacks, as you explain here, why then not to choose another more independent variable to define meaningful basin descriptors?

We have chosen the calibrated parameter values to define basin descriptors, as the main objective is to imitate the model's parameter calibration in gauged regions for ungauged regions. As we estimate one parameter, it is a plausible assumption to look directly for relationships between basin descriptors and the calibrated values (e.g., done by Merz & Blöschl, 2004; Wagener & Wheater, 2006, even for multiple calibration parameters). We revised the text to provide a rationale (and alternatives) for this approach (ll. 180–184) and discussed the low correlation in the light of other studies (ll. 197-202).

Line 177: More than a complex relationship, a very uncertain one, if there is one at all.

Thanks for this opportunity to consider this matter further. To reduce the impact of the (uncertain) descriptor selection in our study, given the low correlation coefficients, we slightly modified our study design to reduce the impact of the correlation-based selection on the tested descriptor sets: Instead of using only the (weakly) correlated climatic and physiographic descriptors in the subsets "cl" and "p", we use now all climatic and physiographic descriptors. This modification enables the assessment of the impact of using only climatic or physiographic descriptors without being influenced by the selection mentioned above (s. ll. 206-215).

Table 1: Define IG(&) in the Table

Thanks for the hint. We now defined it in the table with a [1] (s. ll. 196/197).

Line 210 to 214: Please clarify, the evaluation is done by comparing regionalized gamma with calibrated gamma? Or model predictions? Why a MAE value of 2.1?

Thanks for highlighting that the evaluation method lacked clarity. We now comprehensively emphasize that and why we use the difference between regionalized and calibrated values for evaluation. (ll. 241-253). As stated above, we changed the MAE to logMAE to account for the varying sensitivity of $\gamma$. Furthermore, we added a KGE evaluation to validate the assumption that smaller logMAE values lead to higher discharge accuracy.

Formerly, we added information on an MAE of 2.1 to create a reference for the MAE (comparable to zero for the NSE). We removed such information from the text because it was more confusing than helpful. Instead, we refer to our benchmark-to-beat for evaluation purposes (which is also more in line with the study setup). Thanks!

Line 220 to 222: Please explain better the "tunning approach", what are gamma 1 and gamma 2?

Thanks for highlighting that the description of the tuning required further information. We added information regarding $\gamma_1$ and $\gamma_2$ in the section ll. 229-235. Moreover, restructuring the result section (s. Chapter 3), we have created a result chapter (s. Chapter 3.1) to analyze the effect of tuning more systematically. Here, we also define that using kmeans with three centers on the calibrated $\gamma$ values results in tuning thresholds $\gamma_1 \approx 1.1$ and $\gamma_2 \approx 3.4$ (see ll. 325).

Line 252, why three clusters?

Thanks for the hint that we forgot to mention how we have determined the number of clusters. We have used several indices based on the R-Package nbClust. We have added the information accordingly (ll. 289-292).

Line 259, explain why the "highly flexible version" uses 162 clusters?

We have noted that the "highly flexible version" only indirectly satisfies our goal of analyzing if the clustering beforehand applying knn enhances the estimate. We have now implemented knn directly, where each basin refers to one "cluster", meaning that we used a knn-algorithm that defines the most similar donor basin for each pseudo-ungauged basin (based on Euclidean distance between min-max-normalized attributes). We have revised the text accordingly (ll. 298-301) and adopted all figures in the restructured result section (s. Chapter 3) to evaluate the effect of using knn without clustering beforehand.

Line 305, re-write to clarify.

This sentence is no longer present in the text. Instead of analyzing the boxplots, we now focus on the median logMAE value of the split-sample test ensemble (s. Table 2 in ll. 394/395) and provide the boxplot in Appendix D. The newly introduced table enhances the analysis, providing more detailed insights into the varying basin descriptors' roles. Moreover, we have significantly improved the reliability of our work by sharpening the conditions for the basins used in the study, reducing the number of evaluated basins. We have also slightly modified the study design by using other subsets for 'cl' and 'p' (including all descriptors instead of using only the (weakly) correlated ones).

Line 306 to 308, re-write to clarify the meaning of the sentence. I believe that it has to do more with the relevance of the descriptors included in explaining catchment behavior, or in this case calibrated parameter gamma.

This sentence is no longer present in the text. Instead of analyzing the boxplots, we now focus on the median logMAE value of the split-sample test ensemble (s. Table 2 in ll. 394/395) and provide the boxplot in Appendix D. The newly introduced table enhances the analysis, providing more detailed insights into the varying basin descriptors' roles. Moreover, we have significantly improved the reliability of our work by sharpening the conditions for the basins used in the study, reducing the number of evaluated basins. We have also slightly modified the study design by using other subsets for 'cl' and 'p' (including all descriptors instead of using only the (weakly) correlated ones).

Line 317-317: Change "However" for "Therefore"?, if I understand the intention of the sentence correctly.

Thanks. We have changed 'however' to 'therefore' accordingly (ll. 379).

Line 356-357, the "high flexible" k-means results are not in Figure 4? They need to be added.

We included the results for the newly implemented knn algorithm in all relevant figures in Chapter 3. The algorithm knn is now used instead of the "highly flexible" k-means as the "highly flexible" k-means does not entirely satisfy what we seek to analyze (see above).

Line 358 to 361: I do not understand this statement, why and how a regionalization method will "extract" information form the descriptors. Pagliero et al. (2019) is not correctly cited in the context of this statement.

Due to the changes in our study design (more strictly defined well-behaving basins, other sets of descriptors, and modification of the "highly-flexible" kmeans version), this paragraph is no longer present in the text as (1) we modified the "highly-flexible" version to be more appropriate, and (2) the best variant of the more appropriate "highly-flexible" version leads to comparable results to the best variant of the SI approach. Moreover, when discussing information use, we now avoid the term "extract information", which might give rise to confusion, and prefer "using data more efficiently" instead.

Figure 5d: I have reservations to the validity of comparing global distribution of gamma for WG2 and WG3 given they have different resolutions. Please comment whether this is an issue for this comparison.

We are sorry that it was not clear that the benchmark is not the parameter distribution of WaterGAP2 itself but only the regression equation. This regression equation was fitted to our data and defined as benchmark-to-beat. We have modified the text to clarify it (s. above and ll. 255-263).

Figure 6, what was the reason to choose the year 1989 for this comparison? Seems arbitrary. Same for year 2010 in Figure 7. Why not use an average year, or the driest/wetter year of the period.

Thanks for the suggestion. As we restructured Chapter 3, Figures 6 and 7 changed. We formerly displayed only exemplary years to highlight the local impacts of regionalization. However, we implement the suggestion in the modified figures, as we believe it leads to more meaningful results. In this figure, we now evaluate the variation in run-off induced by varying regionalization methods for the complete simulation period from 1980-2016.

Line 453-454. This statement contradicts itself. Basin descriptors selected for regionalization are of crucial importance because they need to be selected due to the amount of information they contain.

This sentence is no longer present in the text. Instead of analyzing the boxplots, we now focus on the median logMAE value of the split-sample test ensemble (s. Table 2 in ll. 394/395) and provide the boxplot in Appendix D. The newly introduced table enhances the analysis, providing more detailed insights into the varying basin descriptors' roles. Moreover, we have significantly improved the reliability of our work by sharpening the conditions for the basins used in the study, reducing the number of evaluated basins. We have also slightly modified the study design by using other subsets for 'cl' and 'p' (including all descriptors instead of using only the (weakly) correlated ones).

Line 459: use a more academic term instead of "blurring".

This sentence is no longer present in the text due to the restructuring of Chapter 3.

**References**

Batjes, N. H., ISRIC-WISE derived soil properties on a 5 by 5 arc-minutes global grid (ver. 1.2). Report 2012/01, ISRIC – World Soil Information, Wageningen, **[dataset]** accessed on 2020-06-22 from https://data.isric.org/geonetwork/srv/eng/cata-log.search#/metadata/82f3d6b0-a045-4fe2-b960-6d05bc1f37c0, 2012.

Friedl, M., Sulla-Menashe, D.:. MCD12Q1 MODIS/Terra+Aqua Land Cover Type Yearly L3 Global 500m SIN Grid V006,. NASA EOSDIS Land Processes DAAC, **[dataset]** accessed on 2020-11-30 from https://doi.org/10.5067/MODIS/MCD12Q1.006, 2019.

Merz, R., Blöschl, G.: Regionalization of catchment model parameters, Journal of Hydrology, 287, 95-123, https://doi.org/10.1016/j.jhydrol.2003.09.028, 2004.

Wagener, T., & Wheater, H. S.: Parameter estimation and regionalization for continuous rainfall-runoff models including uncertainty, Journal of Hydrology, 320, 132-154, https://doi.org/10.1016/j.jhydrol.2005.07.015, 2006.

---

## Author Comment (AC4)

**Response to Reviewer 2**

We thank the referee for the helpful review and the constructive comments. His/her comments are given below in black, and our responses are in blue. Proposed changes to the manuscript are in red.

This manuscript investigated the ability of parameter transfer to ungauged basins in global hydrological models based on the calibration experiments of WaterGAP3. The combination of traditional and novel regionalization approaches revealed that machine learning-based methods may be too flexible for regionalizing. However, some findings are not convincing and need more discussions or elaboration, some terms need better definitions, and the figures need improvements. Therefore, I recommend a major revision. Please see below for my detailed comments.

Thanks for this comprehensive summary of the critical points. In summary, we restructured Chapter 3 to elaborate more intensely on the findings. By this, we added a KGE evaluation for one representative split-sample test (s. Chapter 3.3) and modified the figures (in Chapters 3.1-3.3) to show more precisely the differences between the methods. Moreover, we extended the analysis of the impacts on the runoff simulations in Chapter 3.4 to (1) gain a deeper understanding of the impacts of regionalization methods in run-off simulations and (2) be more in line with the improved title.

Further, we reduced the set of examined basins to create more valid results. This decision was based on the observation that using only the bias in monthly discharge is inadequate to define a "sufficient model performance" because it does not account for differences in timing or variance. Therefore, we added a minimal KGE in monthly discharge of 0.4 as additional criterion for selecting basins where WaterGAP3 is capable of representing well the hydrological processes (s. ll. 134-139).

Major comments:

1) Title. The term "Regionalization and its impact" is too vague to present the content of this manuscript. Elaborate the titles based on the novelty of the presented work.

Thanks for the suggestion. We adjusted the title to be more precise, highlighting that the study focuses on the regionalization of a GHM and evaluating its impact on runoff simulations. Also, we revised the text in several places to emphasize the novelty of the work, i.e., applying multiple regionalization methods for the first time on the global scale using WaterGAP3, evaluating the methods, and evaluating their impact on the runoff simulation.

Previous title: Regionalization and its impact on global runoff simulations: A case study using the global hydrological model WaterGAP3 (v 1.0.0)

New title: Regionalization *in global hydrological models* and its impact on runoff simulations: A case study using the global hydrological model WaterGAP3 (v 1.0.0)

2) The range of gamma values is too narrow. The cluster of values around the maximum range of gamma implies that the higher values seem appropriate but are excluded from the candidates. Please recalibrate gamma values using wider ranges, and present the scores in the accuracy of predicted streamflow in terms of score index (e.g., NSE).

Thanks for this suggestion. We understand that the parameter range and clustering of the calibrated parameter may be addressed more intensively to make it more comprehensible for peers. Therefore, we address this issue by a brief sensitivity analysis and discussion in the Appendix (see Appendix B in the revised manuscript). In a nutshell, high γ values are primarily sensitive when soil moisture is high and generally less sensitive than lower γ values. High γ values frequently occur in dry regions (see Figure 4b in Müller Schmied et al., 2021). In dry areas, it is not expected that the soil moisture has high values (e.g., see Khosa et al., 2020 and Oloruntoba et al., 2024 for estimated and measured soil moisture in Africa and Draper et al., 2008 for estimated and measured soil moisture in Australia). Therefore, it is not expected that higher gamma values will significantly enhance the calibration result or decrease the issue of clustered calibrated parameter values at the higher end of the parameter space. It is more likely that adding missing model processes, e.g., evaporation from rivers or inaccurate representation of groundwater processes, will solve the clustering of calibrated parameters for dry regions

(Eisner 2015, p. 49). Therefore, we do not change the parameter bounds for γ (please note that the same bounds are also used in Eisner 2015, p. 16; Müller Schmied et al., 2021; Müller Schmied et al., 2023).

As the model performance in calibration is critical when fitting a regionalization method to become valid, we sharpened the criteria for the basins used in this study. Thus, we now exclude all basins with a monthly KGE below 0.4 (933 basins remain after adding this additional criterion) (s. ll. 133-143).

3) The authors assessed the impact of regionalization only from the mean absolute errors in gamma values but neglecting the ability of parameters to represent the river discharges. The parameter transferred from one basin to another should reasonably represent river discharges (accepted behavior) at least in the donor basin. The success of the parameter transfer should be assessed by the accuracies of simulated river discharges in the transferred (ungauged) basins with the parameter. Missing information on the accuracies in the simulated river discharges (both in donor and transferred basins) impedes the reliability of this work.

Thanks for the valuable comment. We agree that including accuracy in simulated river discharge will increase the reliability of the paper. Therefore, we included a new chapter in the result section (see Chapter 3.3), where we added an evaluation of KGE values for one representative split sample.

As the model is very demanding in terms of computational time, conducting the analysis on all split-sample tests was not affordable. However, the additional analysis validates the assumption that logMAE values are a suitable tool for pre-selecting suitable methods for WaterGAP3. Nevertheless, we also recommend in the manuscript adding such an analysis when analyzing the regionalization of WaterGAP3, "as the logMAE of calibrated and regionalized parameter values simplifies the inherent complexity between model parameters and model performance" (ll. 459-461). Please note that we modified the MAE to logMAE to account for the high sensitivity of smaller γ values.

Minor comments:

1) L 174. The usage of 'heavy-tailed' distribution seemed wrong. In probability distribution, heavy-tailed distributions have heavier tails than the exponential distribution. The authors may intend different meanings, but the use of 'heavy-tail' here would misleading.

Thanks for the hint. We modified the term and using know "cluster of calibrated parameter values at the extremes of the valid parameter space" or similar throughout the complete text (e.g., ll. 152, 333).

2) L102. Srmax values are not presented in the manuscript. Maybe it is based on the look-up tables of this model. Please provide it as many researches indicated that soil storage is an important parameter for GHM.

Thanks for the suggestion. We added a global map of the maximal soil moisture to the Appendix (see Appendix A in the revised manuscript) and supplied it in the GitHub repository as .tiff.

3) L259. Explain why 162 groups were used.

We have noted that the "highly flexible version" only indirectly satisfies our goal of analyzing if the clustering beforehand applying knn enhances the estimate. We have now implemented knn directly, where each basin refers to one "cluster", meaning that we used a knn-algorithm that defines the most similar donor basin for each pseudo-ungauged basin (based on Euclidean distance between min-max-normaliued attributes). We have modified the text accordingly (ll. 297-301) and adopted all figures in the restructured result section (s. Chapter 3) to evaluate the effect of using knn without clustering beforehand.

---

## Author Response (AR2)

*We thank the editor for the helpful review and the constructive comments. His comments are given below in bold, and our responses are given in italic. Proposed changes to the manuscript are in* red*. The information about the modified lines is referring to the new version of the manuscript without tracked changes. Additional references are provided at the end of this document.*

**Public justification (visible to the public if the article is accepted and published):**
**There are a few comments to be addressed.**

**In applying the machine learning techniques, could you construct feature importance score bar graph for insights on the influences of the various predictors on the targeted variable or parameter?**

*Thanks for suggesting importance score bars to give insights into the predictors' importance. As using the "p+cl" descriptor set for the tuned MLR version and knn results in the lowest logMAE values but is rather optimal (because it was initially considered as a control group for the selected subset of the descriptors), we use the feature importance bars to give first insights on how to optimize the descriptor sets for these two methods in further studies.* *The two feature importance score bars can be found in Appendix E (ll. 833-839).*

**The use of correlated predictors for the regionalization method comprises the influence of multicollinearity on the study results. There are many ways to avoid multicollinearity in modelling or regionalization. For instance, only one of any two significantly correlated predictors can be considered for modelling or analysis. Therefore, include some discussion on the possible future direction for addressing the issue of multicollinearity in your model or regionalization approach. You may find relevant information from the publication via https://doi.org/10.1002/wrcr.20315 to guide you in the discussion.**

*Thanks for emphasizing the importance of multicollinearity.* *A new section,"Challenges and Future Directions", has been added to the manuscript, in which the issue of multicollinearity is discussed in greater detail (ll. 686-696).* *We suggest several methods for dealing with multicollinearity, namely the use of PCR and PLS to create predictors with low multicollinearity and the explicit checking for multicollinearity in predictor sets using VIF.*

**Provide reference for Equation 1. Provide reference for the Kling-Gupta-Efficiency (KGE). See article published via https://doi.org/10.1016/j.jhydrol.2009.08.003 for the citation. Furthermore, provide the formulae for KGE and logMAE.**

*The two equations and the related references were added to the manuscript (KGE: l. 137, ll. 146-148, logMAE: ll. 258-261).* *Thanks for drawing attention to this lack of completeness.*

**Many methods exist for evaluating model quality. In the discussion, highlight on the point that the choice of an objective function comprises a sub-source of calibration uncertainty. Relevant articles on this can be found via https://doi.org/10.5194/piahs-385-181-2024 and https://doi.org/10.5194/adgeo-5-89-2005 . The idea is that, the use of other methods for model performance evaluation could be recommended for analysis to take into account the influence of the choice of an objective function on modelling results for regionalization.**

*Thanks for emphasizing the role of benchmark selection for model performance evaluation. Two new paragraphs have been added to the end of Section 3.3 in order to elaborate further on this issue (ll. 517-537). Section 3.3 is primarily concerned with the evaluation of the KGE. The newly added paragraphs explain the rationale behind selecting the KGE, emphasizing that the choice is purpose-dependent. Moreover, we created an additional evaluation using a modified version of NSE (as suggested by Krause et al., 2005) and supplied it to the appendix (s. Appendix F, ll. 840-856). Adding this analysis highlights the inherent uncertainty in model evaluation due to the imperfectness of evaluation criteria. Furthermore, it substantiates the value of ensemble runs, which we utilize in Section 3.4, enabling the selection of multiple regionalization methods for the evaluation of their impact on runoff simulations.*

**It is important to elaborate on why many predictors were considered. In fact, a robust regionalization-related model would that with as few predictors as possible. The idea of using many predictors (regardless of the value that each one adds) tends to be a trick to enhance performance of regionalization-related model. This is a misleading procedure. Thus, in such a case, it is recommended to make use of an adjusted R-squared, for instance following Ezekiel (1930) or https://psycnet.apa.org/record/1931-02963-000 to evaluate performance of the regionalization procedure.**

*Thanks for addressing this issue. This statement primarily concerns selecting the descriptor set "p+cl" for MLR tuned and knn. The two other selected methods, SP and SI ensemble ("subset"), are unrelated to this issue as SP uses spatial distances as the metric, and the chosen SI approach uses only five predictors for regionalization. We created two new paragraphs in Section 3.2 to elaborate further on the descriptor selection for the two methods (ll. 422-438) as we consider this issue to be highly relevant for enhancing the study's reliability.*

*In summary: We strongly agree that the use of a reduced number of predictors (or, in general terms, a minimal number of degrees of freedom) should be a fundamental consideration in the construction of models to reduce the risk of over-parametrization, i.e., increasing the model's robustness. Choosing 12 descriptors for tuned MLR and knn, we argue the following.*

(1) *The models remain stable in model quality with no discernible decline from training to testing indicating their robustness. This is evidenced by using 50 % of the basins for training and testing, repsectivley, thus leaving a high proportion of basins for testing (cf. Table D1).*

(2) *Using 12 descriptors is in the order of other studies, even if is in the upper tail (e.g., McIntyre et al., 2013, used three predictors; Beck et al., 2016, used eight predictors; Chaney et al., 2010, used 13 predictors).*

(3) *The initial hypothesis was that the descriptor set "p+cl" would function as a control group to validate the suitability of the selected subset of descriptors. Consequently, the predictor selection for tuned MLR and knn using the descriptor set "p+cl" is not optimal, although it is robust. It would be beneficial to optimize this in future studies, for example, by considering feature importance scores.*

*It should be noted that the use of R-square in place of logMAE would emphasize larger errors, namely those occurring in the upper (less sensitive) tail of the parameter range. Therefore, we omit using the adjusted R-square as an additional metric as it may yield erroneous implications in the context of our study.*

**Identify and elaborate on the general challenges and/or limitations of your study or regionalization. Also highlight on the future directions to guide on how the relevant improvements could be made.**

*Thanks for addressing this issue. A new Section 3.5, "Challenges & Future Directions", has been added to the manuscript in order to address this issue (ll. 662-696). In this chapter, we addressed the considerable runtime required for GHMs, which impedes the execution of extensive regionalization experiments, the predictor selection for regionalization, where no universally valid approach exists and where our study can further be enhanced, and the explicit consideration of multicollinearity in the case of our study. Moreover, we added some potential future directions into the conclusion section (ll. 713-715) and modified the last sentence in the abstract (ll. 28-29).*

*References:*
*Krause, P. Boyle, D. P., Bäse, F.: Comparison of different efficiency criteria for hydrological model assessment, Advances in Geosciences, 5, 89-97, https://doi.org/10.5194/adgeo-5-89-2005, 2005.*

---

## Author Response (AR3)

Dear GMD team,

We compiled a detailed list of all made changes to the accepted manuscript to completely follow the GMD standards and fulfill the journal's high quality. Please note that we did not change the content of the plots, tables, equations, or the manuscripts' text. All changes were due to formatting or to increase the consistency in naming labels/axes (e.g., s. changes in Figure 5). The following changes we made:

- Latitude and longitude information are not used anymore in the global plot in Figure 1a as it is not necessary to understand the plot (as it is showing the globe) → so there is no spacing problem, i.e., we avoid writing 120°N where 120° N would be required after the journal's guideline.
- We added the unit in the y-axis label of Figure 1b ("Count" becomes "Count (-)")
- In Figure 2, we renamed "Catchment" to "Basin" in two cases to be consistent with the rest of the manuscript.
- The naming of gamma is now consistent in all plots (using the Greek symbol instead of "gamma"); this affects Figure B1 and Figure 5.
- The legend in Figure C1 is not cut anymore.
- The spacing in Figure C2 is enhanced so numbers on the x-axis can be read more easily and units are added.
- The naming in Figure D1 changed so that "WG2" became "B2B" (to be consistent with all other plots).
- The x-axis labels and the y-axis in Figure E1 are now fully shown.
- The y-axis label in Figure F1 is now "modified NSE (-)" instead of "NSE_1 (-)" to be consistent with the naming in the text.
- All composite figures are now consistently produced by merging the *.png files using PowerPoint and exporting them in high resolution. Letters indicating sub-figures ((a), (b), …) are written consistently in Arial.
- The first letter of all axis labels is now consistently written as a capital letter.
- As we made minor changes to the code for plotting the results, we updated our GitHub repository and included the newly generated DOI from the new release (https://zenodo.org/records/13122859).
- We moved the *Code and data availability*, *Authors contribution*, and *Competing interests* statements below the Appendix following the GMD guideline.
- We deleted the page breaks between the Appendices.
- We removed a double comma in Appendix B.